

# An Offline Framework for High-dimensional Ensemble Kalman Filters to Reduce the Time-to-solution

Yongjun Zheng, Clément Albergel, Simon Munier, Bertrand Bonan, and Jean-Christophe Calvet

CNRM, Université de Toulouse, Météo-France, CNRS, Toulouse, France

**Correspondence:** Y. Zheng (zhengyongjun@gmail.com)

**Abstract.** The high computational resources and the time-consuming IO (Input/Output) are major issues in offline ensemble-based high-dimentional data assimilation systems. Bearing these in mind, this study proposes a sophisticated dynamically running job scheme as well as an innovative parallel IO algorithm to reduce the time-to-solution of an offline framework for high-dimensional ensemble Kalman filters. The dynamically running job scheme runs as many tasks as possible within a single job to reduce the queuing time and minimize the overhead of starting/ending a job. The parallel IO algorithm reads or writes non-overlapping segments of multiple files with an identical structure to reduce the IO times by minimizing the IO competitions and maximizing the overlapping of the MPI (Message Passing Interface) communications with the IO operations. Results based on sensitive experiments shown that the proposed parallel IO algorithm can significantly reduce the IO times and has a very good scalability, too. Based on these two advanced techniques, the offline and online modes of ensemble Kalman filters are built based on PDAF (Parallel Data Assimilation Framework) to comprehensively assess their efficiencies. It can be seen from the comparisons between the offline and online modes that the IO time only accounts for a small fraction of the total time with the proposed parallel IO algorithm. The queuing time might be less than the running time in a low-loaded supercomputer such as in an operational context but the offline mode can be nearly as fast as, if not faster than, the online mode in terms of time-to-solution. However, the queuing time is dominant and several times larger than the running time in a high-loaded supercomputer. Thus, the offline mode is substantially faster than the online mode in terms of time-to-solution, especially for large-scale assimilation problems. From this point of view, it suggests that an offline ensemble Kalman filter with an efficient implementation and a high performance parallel file system should be preferred over its online counterpart for the intermittent data assimilation in many situations.

## 1 Introduction

Both the numerical model of a dynamical system and its initial condition are imperfect owing to the inaccuracy and incompleteness to represent the underlying dynamics and to measure its states. Thus, to improve the forecast of a numerical model,



data assimilation (DA) methods combine the observations and the prior states of a system to estimate the posterior states (usually more accurate) of the system taking into account their uncertainties. Two well-known DA methods are the variational technique and the ensemble-based technique. The hybrid methods combining the advantages of the variational technique and the ensemble-based technique have gained increasing interest in recent years. Bannister (2017) gives a comprehensive review
of variational, ensemble-based, and hybrid DA methods used in operational contexts.

The ensemble-based methods not only estimate the posterior state using the flow-dependent covariance but also practically compute the uncertainty of the estimation. The Kalman filter is an unbiased optimal estimator for a linear system (Kalman, 1960). The extended Kalman filter (EKF) is a generalization of the classic Kalman filter to a nonlinear system. It uses the tangent linear models of the nonlinear dynamical model and the nonlinear observation operators to explicitly propagate the
probability moments. For a high-dimensional system, the explicit propagation of the covariance is almost infeasible. The ensemble Kalman filter (EnKF) is an attractive alternative to the EKF. It implicitly propagates the covariance by the integration of an ensemble of the nonlinear dynamical model that makes its implementation simple owing to the elimination of the tangent linear model. Since the introduction of the EnKF by Evensen (1994), many variants of the EnKF have been proposed to improve the analysis quality or the computational efficiency. For example, the stochastic ensemble Kalman filter perturbs
the observation innovation to correct the premature reduction in the ensemble spread (Burgers et al., 1998; Houtekamer and Michell, 1998); the ensemble square room filter (EnSRF) introduced the square root formulation to avoid the perturbations of the observation innovation (Whitaker and Hamill, 2002; Sakov and Oke, 2008); the ensemble transform Kalman filter (ETKF) explicitly transforms the ensemble to obtain the correct spread of the analysis ensemble (Bishop et al., 2001) and the local ensemble transform Kalman filter (LETKF) is widespreadly adopted owing to its efficient parallelization (Hunt et al., 2007);
and the error subspace transform Kalman filter (ESTKF, Nerger et al. (2012a)) and its localized variant (LESTKF) combine the advantages of the ETKF and the singular evolutive interpolated Kalman filter (SEIK, Pham et al. (1998)). For comprehensive reviews of the EnKF, we refer the readers to the ones by Vetra-Carvalho et al. (2018) and Houtekamer and Zhang (2016).

Many schemes have been proposed to reduce the computational cost of the EnKF, especially to reduce the computational cost of the large matrix inverse or factorization. Keppenne (2000) used a domain decomposition to perform the analysis on
distributed-memory architectures to avoid the large memory required by the entire state vectors of all the ensemble members. The sequential method assimilates one observation at a time (Cohn and Parrish, 1991; Anderson, 2001) or multiple observations in each batch (Houtekamer and Mitchell, 2001). The LETKF decomposes the global analysis domain into local domains where the analysis is computed independently (Hunt et al., 2007). Godinez and Moulton (2012) derived a matrix-free algorithm for the EnKF and showed that it is more efficient than the singular value decomposition (SVD) based algorithms. Houtekamer
et al. (2014) gave a comprehensive description of the parallel implementation of the stochastic EnKF in operation at the Canadian Meteorological Centre (CMC) and pointed out the potential computational challenges. When the sequential technique associates with the localization, the analysis is suboptimal and dependent on the order of observations (Nerger, 2015; Bishop et al., 2015). Steward et al. (2017) assimilated all the observations simultaneously and directly solved the large eigenvalue problem using the Scalable Library for Eigenvalue Problem Computations (SLEPc, Hernandez et al. (2005)).





As mentioned by Houtekamer et al. (2014), an EnKF system has to efficiently use the computer resources, such as disk space, processors, main computer memory, memory caches, job-queuing system, and archiving system, in both research and operational contexts to reduce the time-to-solution. To obtain a solution, the EnKF system has to perform a series of tasks including the observations preprocessing, the jobs queuing, ensemble members running, the analysis, the post-processing, the

archiving, and so on. Thus, the time-to-solution is the total time to obtain a solution, that is, the time from the beginning to the end of an experiment, such as one assimilation cycle in operational context or ten-year reanalyses in a research context. Even with the efforts of the aforementioned literature, the time-to-solution of an EnKF system is still demanding. For instance, the global land data assimilation system (LDAS-Monde, Albergel et al. (2017)) uses an SEKF (Simplified Extended Kalman Filter, Mahfouf et al. (2009)) or an EnKF scheme (Fairbairn et al., 2015) to assimilate satellite-derived terrestrial variables in the

Interactions between Soil, Biosphere, and Atmposhere (ISBA) land surface model within the Surface Externalisée (SURFEX) modelling platform (Masson et al., 2013). By assimilating satellite-derived terrestrial variables, LDAS-Monde improves high spatial-temporal resolution analyses and simulations of land surface conditions to extend our capabilities for climate change adaptions. But at a global scale or even at a regional scale with a high spatial resolution (1km x 1km or finer), it becomes challenging in terms of time-to-solution. This is the motivation of the comprehensive evaluations of different implementations

of an EnKF system to determine which technique should be adopted for an efficient and scalable framework for LDAS-Monde.

There are two modes to implement an EnKF: offline and online modes. The offline mode is the most extensively adopted strategy, especially in the operational context of numerical weather prediction (NWP) where the operational DA process is intermittent and consists of an alternating sequence of short-range forecasts and analyses. In offline mode, the dynamical model and the EnKF are totally independent, that is, these two components are two separate systems. An ensemble of the

dynamical model runs until the end of the cycle and outputs the restart files and stops; then the EnKF system reads the ensemble restart files and observations to produce the analysis ensemble which update the restart files, also output the analysis mean (the optimal estimation of the states, see Figure 1). Traditionally, the dynamical model and the DA system are developed separately. The offline mode keeps the independence of these two systems which is highly desirable for each community. Thus, the implementation and maintenance of an offline mode is simple and flexible. One big disadvantage of an offline mode is

its time-consuming IO (Input/Output) operations, especially for a high-dimensional system and a large number of ensemble members. Recently, several online modes have been proposed to avoid the expensive IO operations of the offline mode (Nerger and Hiller, 2013; Browne and Wilson, 2015). The online mode forms a coupled system of the dynamical model and the EnKF which exchange the prior and posterior states by message passing interface (MPI) communications. When observations are available, the MPI tasks of dynamical models send their forecast ensemble members (prior states) to those of the EnKF, then

the MPI tasks of the EnKF combine the observations and the received forecast ensemble members (prior states) to generate and send back the analysis ensemble members (posterior states), then the MPI tasks of dynamical models resume their running. The development of a coupled system demands a substantial time and effort. Another disadvantage of the online mode is the large job-queuing time because running the ensemble simultaneously requires a large number of nodes when both the number of ensemble members and the number of nodes per member are large. With the consideration of possible prohibitive IO operations

for an offline EnKF, the online frameworks proposed in the literature seem promising and were claimed to be efficient (Nerger



and Hiller, 2013; Browne and Wilson, 2015). But to our best knowledge, there have been no attempts to assess the time-to-solution of an offline EnKF against that of an online EnKF. In this context, our study tries to answer the next questions: Is an online EnKF really faster than an offline EnKF? Can an offline EnKF be as fast as, if not faster than, an online EnKF with a good framework and algorithms using advanced techniques of parallel IO?

An offline EnKF system simultaneously submits the jobs (usually one ensemble member per job) to the supercomputer. With high priority as in an operational context, all of the jobs might get run immediately, and this is the most efficient way. But in a research context, each job usually needs to wait in the job queue for a period before it gets run. Sometimes, the job-queuing time is significantly larger than the actual running time in a high-loaded machine if the job requires a large number of computer nodes or a long running time. In addition, the resource management and scheduling system of a supercomputer needs time to

allocate the required nodes for a job, start and stop the job; these overheads are not negligible. It is then desirable to minimize the impact of the job queuing and overheads. This is the first object of this study to reduce the time-to-solution of an offline EnKF.

    Massive IO operations pose a great challenge on the implementation of an offline EnKF system for high-dimensional assimilation problems. Yashiro et al. (2016) presented a framework with a novel parallel IO scheme for the NICAM (Nonhydrostatic

ICosahedral Atmospheric Model) LETKF system. This method uses the local disk of the computer node and only works for architectures with a local disk of large capacity in each computer node. Nowadays, most supercomputers have parallel file systems. With the progress of technologies in high performance computing (HPC), the state-of-art parallel file system has an increasingly high-scalability, high-performance, and high-availability. Several parallel IO libraries based on PnetCDF (Parallel netCDF project, 2018) or netCDF (Unidata, 2018) with parallel HDF5 (The HDF Group, 2018) have been developed for NWP

models and climate models. XIOS (ISPL, 2018) can read and write in parallel but cannot update variables in a netCDF file. CDI-PIO (DKRZ and MPI-M, 2018) and CFIO (Huang et al., 2014) can only write in parallel. PIO (NCAR, 2018) is very flexible but is not targeted for the offline EnKF system which synchronously reads then updates multiple files with an identical structure. Thus, with advanced parallel IO techniques and innovative algorithms, the second object of our work to reduce the time-to-solution is to answer the following question: Can the IO time of an offline EnKF be a negligible fraction of the total

time?

    To address the aforementioned challenges of an offline EnKF, we propose a sophisticated dynamically running job scheme and an innovative parallel IO algorithm to reduce the time-to-solution, and comprehensively compare the time-to-solutions of the offline and online EnKF implementations. This paper is organized as follows. The formulation of an EnKF, its parallel domain decomposition method, an offline EnKF, and an online EnKF are described in section 2. The sophisticated dynam-

ically running job scheme aiming to minimize the job queuing and overheads and the innovative parallel IO algorithm are detailed in section 3. The experimental environments, designs, and the corresponding results are presented in section 4. Finally, conclusions are drawn in section 5.



## 2 Ensemble Kalman Filters

In an EnKF, each member is a particular realization of the possible model trajectories. Assuming there are $N_e$ ensemble members $\boldsymbol{x}_1, \cdots, \boldsymbol{x}_{N_e}$, where the subscript denotes the member ID, $\boldsymbol{x} \in \mathcal{R}^{N_x}$ is the state vector, and $N_x$ is the dimension of state space. let $\mathbf{X} = [\boldsymbol{x}_1, \cdots, \boldsymbol{x}_{N_e}] \in \mathcal{R}^{N_x \times N_e}$ be the ensemble matrix, thus, the ensemble mean is

$$\overline{\boldsymbol{x}} = \frac{1}{N_e} \sum_{k=1}^{N_e} \boldsymbol{x}_k, \tag{1}$$

the ensemble perturbation matrix is

$$\mathbf{X}' = [\boldsymbol{x}_1 - \overline{\boldsymbol{x}}, \cdots, \boldsymbol{x}_{N_e} - \overline{\boldsymbol{x}}], \tag{2}$$

and the ensemble covariance matrix is

$$\mathbf{P} = \frac{\mathbf{X}'\mathbf{X}'^T}{N_e - 1}. \tag{3}$$

Further, let $\boldsymbol{d} = \boldsymbol{y} - \mathcal{H}(\boldsymbol{x})$ be the innovation vector, where $\boldsymbol{y} \in \mathcal{R}^{N_y}$ is the observation vector, $\mathcal{H} : \mathcal{R}^{N_x} \to \mathcal{R}^{R_y}$ is the nonlinear observation operator which maps the state space to the observation space, $N_y$ is the dimension of observation space.

The Kalman update equation for the state is

$$\boldsymbol{x}^a = \boldsymbol{x}^f + \mathbf{K} \left( \boldsymbol{y} - \mathcal{H}(\boldsymbol{x}^f) \right) = \boldsymbol{x}^f + \mathbf{K}\boldsymbol{d}, \tag{4}$$

and the Kalman update equation for the covariance is

$$\mathbf{P}^a = (\mathbf{I} - \mathbf{KH}) \, \mathbf{P}^f, \tag{5}$$

where the Kalman gain is

$$\mathbf{K} = \mathbf{P}^f \mathbf{H}^T (\mathbf{H} \mathbf{P}^f \mathbf{H}^T + \mathbf{R})^{-1}. \tag{6}$$

Within the above equations, $\mathbf{H}$ is the linear observation operator of $\mathcal{H}$, and $\mathbf{R} \in \mathcal{R}^{N_y \times N_y}$ is the observation error covariance matrix. The superscript $f$ and $a$ denote forecast and analysis, respectively, and the superscript $T$ denotes a matrix transposition.

Using the covariance update equation (5) and the Kalman gain (6), the equation (3) can be written as

$$\begin{aligned}
\mathbf{X}'^a \mathbf{X}'^{aT} &= (N_e - 1)\mathbf{P}^a \\
&= \left(\mathbf{I} - \mathbf{P}^f \mathbf{H}^T (\mathbf{H}\mathbf{P}^f\mathbf{H}^T + \mathbf{R})^{-1}\mathbf{H}\right) \mathbf{X}'^f \mathbf{X}'^{fT} \\
&= \mathbf{X}'^f \left(\mathbf{I} - \mathbf{S}^T \mathbf{F}^{-1}\mathbf{S}\right) \mathbf{X}'^{fT} \\
&= \mathbf{X}'^f \left(\mathbf{W}\mathbf{W}^T\right) \mathbf{X}'^{fT} = \left(\mathbf{X}'^f \mathbf{W}\right) \left(\mathbf{X}'^f \mathbf{W}\right)^T ,
\end{aligned} \tag{7}$$

where $\mathbf{S} = \mathbf{H}\mathbf{X}'^f$, $\mathbf{F} = \mathbf{S}\mathbf{S}^T + (N_e - 1)\mathbf{R}$, and $\mathbf{W}$ is the square root of $\mathbf{I} - \mathbf{S}^T \mathbf{F}^{-1}\mathbf{S}$.





Thus, without explicit computation of the covariances $\mathbf{P}^f$ and $\mathbf{P}^a$, the analysis ensemble can be computed as

$$\mathbf{X}^a = [\overline{\boldsymbol{x}^a}, \cdots, \overline{\boldsymbol{x}^a}] + \mathbf{X}'^f \mathbf{W}, \tag{8}$$

where the analysis mean is

$$
\begin{aligned}
\overline{\boldsymbol{x}^a} &= \overline{\boldsymbol{x}^f} + \mathbf{K}\boldsymbol{d} \\
&= \overline{\boldsymbol{x}^f} + \mathbf{P}^f \mathbf{H}^T (\mathbf{H}\mathbf{P}^f\mathbf{H}^T + \mathbf{R})^{-1}\boldsymbol{d} \\
&= \overline{\boldsymbol{x}^f} + \mathbf{X}'^f \mathbf{S}^T \mathbf{F}^{-1}\boldsymbol{d}
\end{aligned} \tag{9}
$$

by combining the state update equation (4) with the Kalman gain (6).

For most ensemble-based Kalman filters (Burgers et al., 1998; Pham et al., 1998; Houtekamer and Mitchell, 2001; Bishop et al., 2001; Anderson, 2001; Whitaker and Hamill, 2002; Evensen, 2003; Hunt et al., 2007; Livings et al., 2008; Sakov and Oke, 2008; Nerger et al., 2012a), the analysis update can be written as a linear transformation in (8). However, the different variants of ensemble-based Kalman filters use different ways to calculate the tranformation matrix $\mathbf{W}$ which is not necessary

to be the square root as in (7). From the above derivation, it can be seen that the most computationally expensive part is the computation of the square root which involves the inverse of the matrix $\mathbf{F}$. In general, the square root $\mathbf{W}$ can be obtained by a Cholesky decomposition or a singular value decomposition (SVD). The Cholesky decomposition is more efficient than the SVD, but the SVD is more robust if the matrix is significantly ill-conditioned.

## 2.1   Domain Decomposition for Parallel EnKFs

For a high-dimensional system, the size of the state vector $\boldsymbol{x}_k$ is large, therefore it is not practical to perform the EnKF analysis without parallelization. The straightforward way of parallelization is to decompose the state vector $\boldsymbol{x}_k$ into approximately equal parts by $N_{mpi}$ MPI tasks. Because all member state vectors have an identical structure, each member state vector is decomposed in an identical manner, and each member is one column of the ensemble matrix $\mathbf{X}$. Thus, each MPI task computes at most $\lceil \frac{N_x}{N_{mpi}} \rceil$ consecutive rows of the ensemble matrix $\mathbf{X}$. Figure 6 illustrates this decomposition. Each level of a three-dimensional

variable is decomposed in the same way as if a horizontal domain decomposition was used. For multiple variables, the same decomposition is applied to each variable. This domain decomposition has the advantage of a good load balance. Without loss of generality, the descriptions in this study assume the state vector $\boldsymbol{x}_k$ is a one-dimensional variable as a multi-dimensional variable can be viewed as linear in the memory. The domain decomposition is the foundation for the innovative parallel IO algorithm proposed in Section 3.2.2.

## 25   2.2   An Offline EnKF System

An offline EnKF system is a sophisticated system consisting of many components. Figure 1 illustrates the typical workflow of an ensemble-based DA system with its essential components. In an operational context of NWP, a notable feature of an intermittent DA system is the alternating sequence of short-range forecasts and analyses. Each short-range forecast and analysis





forms a cycle. At the beginning of each cycle, All the forecast members read in their corresponding analysis members from last cycle and integrate independently for the period of the cycle and this is called a forecast phase. Usually, each forecast member uses the same dynamical model but with a differently perturbed initial condition, a differently perturbed forcing, or a different set of parameters. Meanwhile, a deterministic forecast is usually integrated for a period longer than the cycle and outputs the

history files more frequently. At the end of the cycle, all the forecast members output their restart files and stop. Then, the EnKF combines the observations and the forecast ensemble ($\boldsymbol{x}_k^f$, or its equivalence $\boldsymbol{A}_k^f$ in Figure 1) to produce the analysis ensemble ($\boldsymbol{x}_k^a$, or its equivalence $\boldsymbol{A}_k^a$ in Figure 1) and the analysis mean ($\overline{\boldsymbol{x}^a}$, or its equivalence $\overline{\boldsymbol{A}^a}$ in Figure 1) , which updates the restart files of the ensemble forecasts and the deterministic forecast, respectively. This is called the analysis phase. This process is repeated for next cycle.

There are several advantages to have an extra deterministic forecast. First of all, the deterministic forecast with the optimal initial condition is integrated over a much longer period than that of the cycle and outputs the history files more frequently which are the user-end deterministic prediction products; this is essential in an operational NWP context. Secondly, the ensemble forecasts only output restart files at the end of the cycle which significantly reduce the IO time and the required disk space. Thirdly, it is even possible to use the deterministic forecast as a member (Schraff et al., 2016).

A distinctive feature of an offline EnKF system is that each ensemble member run is completely independent of each other and the DA component runs only after all the members are run. Each member run has its own queuing time and overheads owing to the involvement of a job system. Because all the member runs have finished when the DA component begins to run, the practically possible way to exchange information between the model component and the DA component is via the intermediate restart files. Reading and writing many restart files whose size is large is time-consuming and may counteract the

simplicity and flexibility of the favorite offline EnKF system. It is very common that the DA system submits one member per job or even a fixed number of multiple members per job. The DA component reads or writes the ensemble restart files with one IO task or as many IO tasks as the ensemble members. This is not efficient, and both these two aspects will be addressed in section 3 accordingly.

### 2.3 An Online EnKF System

As already mentioned, there are several methods to build an online EnKF system. The methods used in this paper is similar to one possible implementation suggested in the parallel data assimilation framework (PDAF, 2018). With the operational NWP in mind, the online EnKF system presented in this paper is also an intermittent DA system. In this system, the model component reads in the ensemble analyses from last cycle and integrates simultaneously $N_e$ ensemble members for the period of a cycle, then scatters the ensemble state $\mathbf{X}^f$ to the DA component which performs the analysis and outputs the ensemble analysis $\mathbf{X}^a$

and ensemble analysis mean $\boldsymbol{x}^a$. Finally the system stops and only restarts in a proper further time for the next cycle. Thus, the main difference between this online EnKF and the offline EnKF described in Section 2.2 is that there is no intermediate outputs, which eliminate the ensemble-writing operations in the model component and the ensemble-reading operations in the DA component, between the forecast and the analysis phases. This effectively reduces the IO operations to half compared to





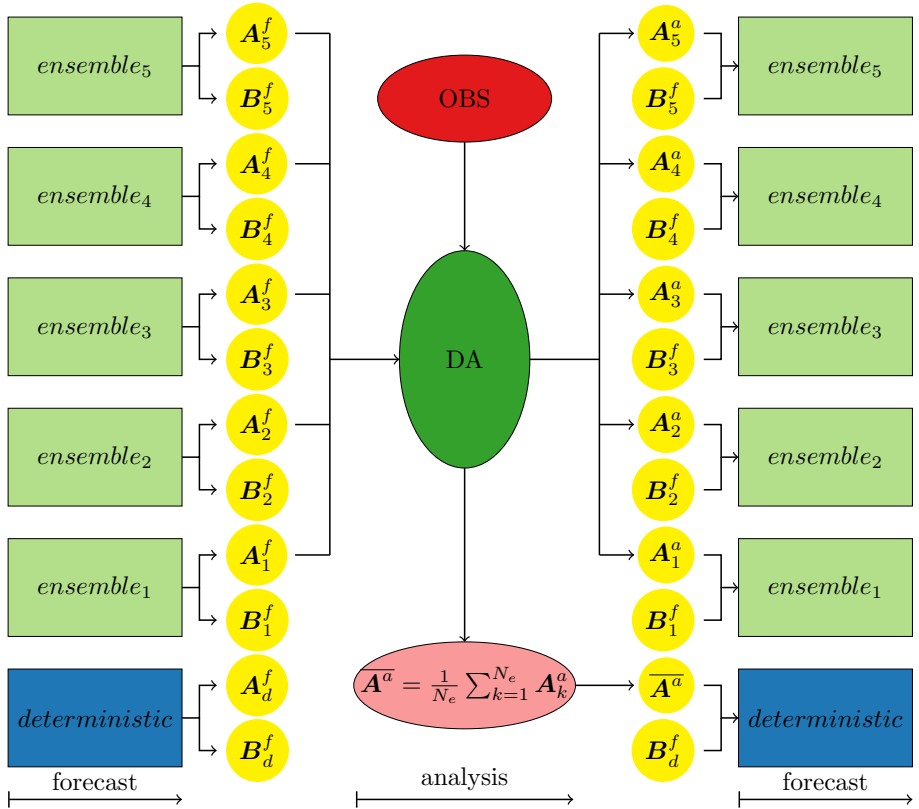

**Figure 1.** Schematic diagram illustrating the workflow of an offline EnKF system. Assuming the forecast model is a coupled model consisting of two components $\boldsymbol{A}$ and $\boldsymbol{B}$, each component outputs its own results $\boldsymbol{A}_k^f$ and $\boldsymbol{B}_k^f$, respectively. But the EnKF system only analyses and updates the state of the component $\boldsymbol{A}$, and only output the analysis ensemble $\boldsymbol{A}_1^a \ldots \boldsymbol{A}_{N_e}^a$ and the analysis mean $\overline{\boldsymbol{A}^a}$. In this example, there are $N_e = 5$ ensemble members which runs $h_e$ hours for each cycle and only outputs the restart files at the end of the cycle. In addition, there is a deterministic forecast started with the optimal initial condition $\overline{\boldsymbol{A}^a}$ from the EnKF system, and this deterministic forecast may run longer than the period ($h_e$ hours) of one cycle and may output more frequently.

the offline EnKF. However, being an intermittent DA system, for each cycle it still needs to read the analysis ensemble from the last cycle and write the analysis ensemble of the current cycle.

Figure 2 illustrates our implementation of the online EnKF system used in this study. In this example, the online EnKF system uses 18 MPI tasks to integrate simultaneously six ensemble members with three MPI tasks per member. The grid cells with the same background color belong to one ensemble member. The numbers to the left of the model column (the tallest column) are the ranks of the MPI tasks in the global MPI communicator. The numbers with a yellow circle inside the model column are the ranks of the MPI tasks in its model MPI communicator of the corresponding member. As shown in the data assimilation column, the first three global MPI tasks also form a filter MPI communicator. This filter MPI communicator is used to perform the EnKF analysis. All the MPI tasks with the same rank (number with a yellow circle) in the model MPI





communicators form a coupled MPI communicator to exchange data between ensemble members and the EnKF component. Thus, every model member has an identical domain decomposition, and so does the DA component. This facilitates the data exchange between ensemble members and the EnKF component. As in Figure 2, each member uses its own three MPI tasks to read in the corresponding initial condition and integrate the model to the end of the cycle; then the first MPI task of each

member sends its corresponding segment of its states to the corresponding row and column of the ensemble matrix $\mathbf{X}$ in the first MPI task of the DA component which, in fact, is the first MPI task of the first member, so do the second and third MPI tasks. Finally the DA component has all the data in the ensemble matrix $\mathbf{X}$ to perform the assimilation analysis and writes out the analysis ensemble as well as the analysis mean. This online EnKF has the disadvantage of wasting computational resources because the MPI tasks starting from the second member are idle when the DA component is running. But it complicates the

data exchange between the model members and the DA component if all the MPI tasks are used for the DA component, also it might not always help to have more MPI tasks for accelerating the assimilation analysis because the scale of the problem determines the number of MPI tasks, and sometimes more MPI tasks might undermine the efficiency of a problem owing to the expensive MPI communications.

## 3 Methods

This section lengthily presents the following two methods in this study to reduce the time-to-solution of an offline EnKF.

### 3.1 Dynamically Running Job Scheme for Minimizing the Job Queuing and Overheads

Using embarrassingly parallel strategy, the jobs of all the members are submitted simultaneously. On a high-loaded machine, each job needs to wait for a long time before running, especially when the job requires a large number of nodes. To reduce the job queuing and overheads, we propose a sophisticated running job scheme to dynamically run the ensemble members

over multiple jobs, as illustrated by Figure 3. First of all, the scheme generates a to-do list file with all the IDs (Identities) of the ensemble members followed by the ID ($= N_e + 1$) of the DA component; then simultaneously submits $N_j$ jobs where $N_j \in [1, N_e]$ can be fewer than the number of members. Because the ID of the DA component is at the end of the to-do list, the proposed scheme automatically guarantees that the sucessful completion of all the members is checked and confirmed before executing the DA component. When a job (for example, $job_1$ in Figure 3) is dispatched to start its running, the job locks the

to-do list file to obtain a member ID (for example, member ① in Figure 3), removes the member ID from the to-do list file and unlocks the to-do list file, then starts the execution of that member. While the job is running (for example, $job_1$ in Figure 3), another job (for example, $job_2$ in Figure 3) gets the required nodes to start its running, obtains a member ID (for example, member ② in Figure 3) and then starts the execution of that member in the same manner. When a job (for example, $job_1$ in Figure 3) finishes the execution of a member (for example, member ① in Figure 3), instead of being terminated, the job

continues to obtain another member ID (for example, member ④ in Figure 3) from the to-do list file then starts the execution of that member. The process is repeated until the to-do list file is empty. The mechanism to lock and unlock the to-do list file is essential to prevent the same member from being executed by multiple jobs.

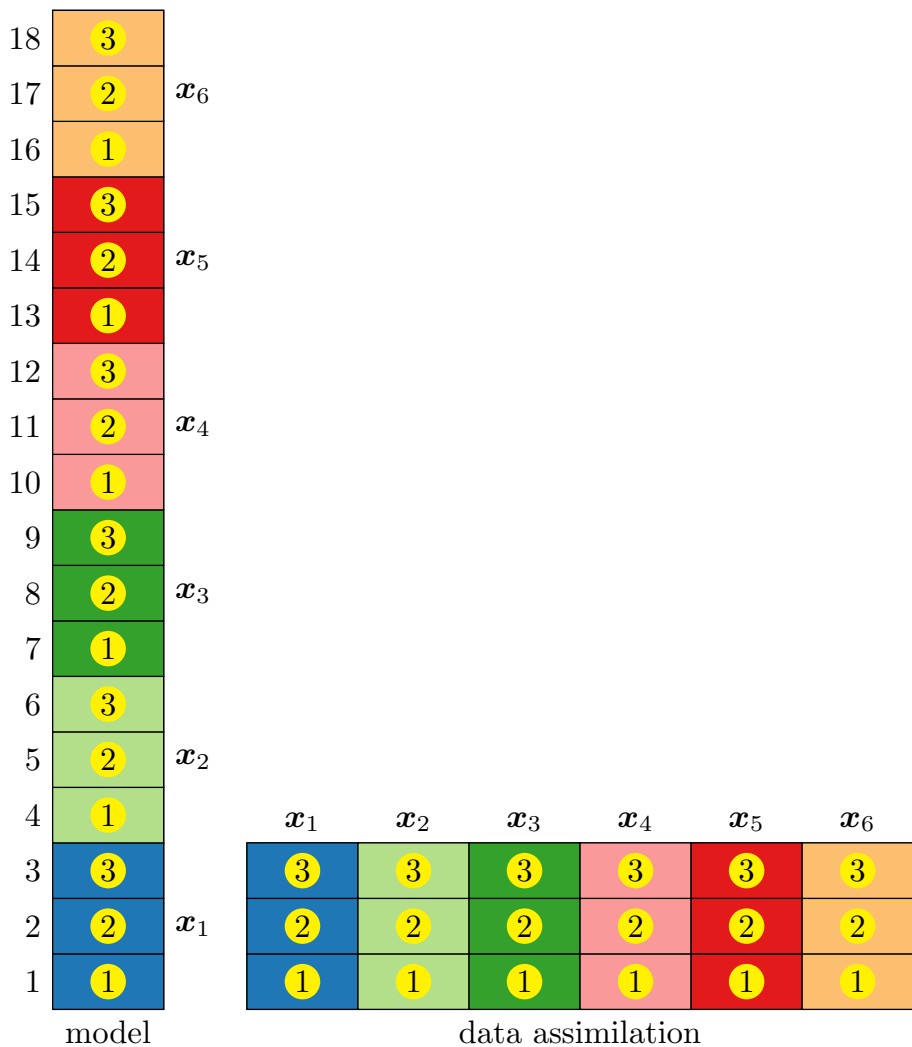

**Figure 2.** Schematic diagram illustrating the implementation of an online EnKF system. In this example, 18 MPI tasks are used to integrate six ensemble members with three MPI tasks per member, then the first three MPI tasks are used to perform the assimilation analysis. The numbers on the left of the model panel (the left panel) are the ranks of the MPI tasks in the global MPI communicator. The grid cells with the same background color in the model panel belong to one ensemble member $x_k$. The numbers with a yellow circle, which demonstrate the parallel domain decomposition of the corresponding member, are the ranks of the MPI tasks in the model MPI communicator of the corresponding member. The data assimilation panel (the right panel) demonstrates how the member state $x_k$ are assembled into the ensemble matrix $\mathbf{X}$ while keeping the same domain decomposition as do the model members.





In most setting of resource management and scheduling systems, the shorter the run time requested by a job is, the shorter the queuing time is. The proposed scheme can specify a time limit of jobs to balance the queuing and the overheads. With a short time limit but not shorter than the execution of a member or the DA component, a job reaches its time limit and the executing member is interrupted. In this case, the ID of the interrupted member is inserted into the front of the to-do list so

that the remaining running jobs can restart the execution of the interrupted member. By carefully tuning the time limit of jobs, interruptions can be minimized. With this sophisticated scheme which dynamically runs the members, sometimes the first several jobs have finished the executions of all the members and the DA component, the remainning jobs are still waiting in the queue and need to be canceled (for example, the $job_5$ in Figure 3 will be automatically canceled after the finish of the DA component in $job_3$). Thus, this scheme substantially reduces the job queuing and overheads.

## 3.2  Parallel IO Algorithm for Improving the IO Performance

### 3.2.1  Lustre Parallel File System

The parallel file system is a crucial component in a current supercomputer. There are several parallel file systems. Lustre parallel file system (http://lustre.org, last access: 18 December 2018) is best known for powering many of the largest HPC clusters worldwide owing to its scalability and performance. The Lustre parallel file system consists of five key components

(see Figure 4). The metadata servers (MDS) make metadata (such as filenames, directories, permission and file layout) available to Lustre clients. The metadata targets (MDT) store metadata and usually use solid-state disks (SSD) to accelerate the metadata requests. The object storage servers (OSS) provide file IO services and network requests. The object storage targets (OST) are the actual storage media where user file data are stored. The file data is divided into multiple objects which are stored on a separate OST. Lustre clients are computational, visualization, or desktop nodes that are running Lustre client software

and mount the Lustre file system. The interactive users or MPI tasks make requests to open, close, read, or write files and the requests are forwarded via an HPC interconnect to the MDS or OSS which performs the actual operations.

The high performance of Lustre file system is mainly attributed to its ability to stripe data across multiple OSTs in a round-robin fashion. Figure 5 illustrates how a file is striped across multiple OSTs. A file is divided into multiple segments of the same size, usually, the last segment is incomplete. The size of each segment can be specified by the stripe size (denoted as

"size" in Figure 5) parameter when the file is created. Similarly, the stripe count (denoted as "count" in Figure 5) parameter is the number of OSTs where the file is stored and can be specified when the file is created. The parameters have default values unless specified explicitly and cannot be changed after the creation of a file. In Figure 5, the file is divided into 13 segments and the stripe count parameter is equal to 5. The first segment goes to the first OST, $\cdots$, the fifth segment goes to the fifth OST which is the last OST of this file; then, the sixth segment goes to the first OST, $\cdots$, repeat this pattern until the last segment.

The optimal stripe parameters usually depend on the file size, the access pattern, and the underlying architecture of the Lustre file system. The stripe size parameter must be a multiple of the page size and using a large stripe size can improve performance when accessing a very large file. Because of the maximum size that can be stored on the MDT, a file can only be striped over



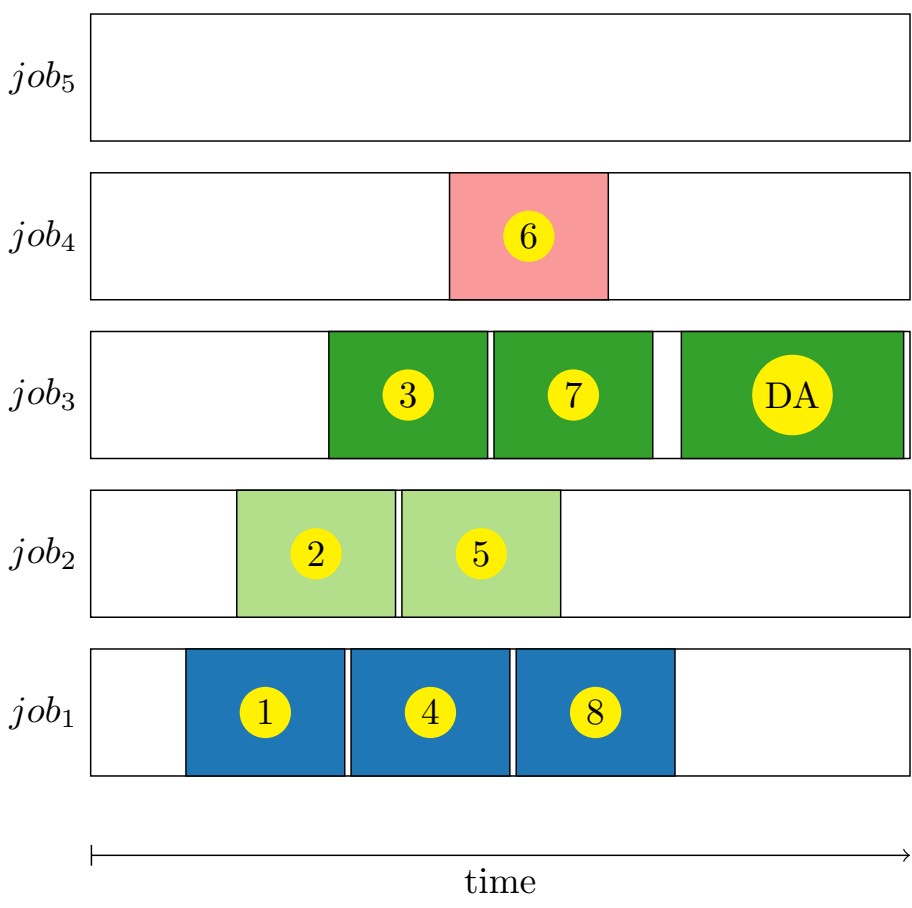

**Figure 3.** Schematic diagram illustrates one possible scenario of the runnings of $N_e$ ensemble members and DA with $N_j$ jobs ($N_e = 8$ and $N_j = 5$ in this example). The number (DA, also) with a yellow circle is the ID of a member (data assimilation), its surrounding colorful grid cell denotes the duration of the running of this member (data assimilation). The blanks before the first colorful grid cell, between the colorful grid cells, and after the last colorful grid cell in each job are the queuing time, the overhead time between two consecutive runnings in a job, and the idle time, respectively.





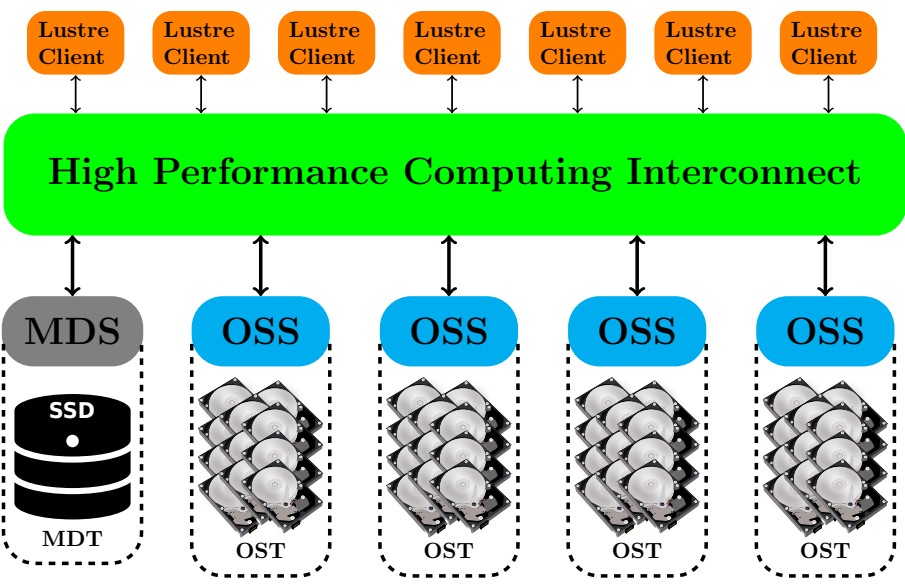

**Figure 4.** Schematic diagram of a Lustre parallel file system, see Section 3.2.1 for the definitions of the abbreviations of "MDS", "OSS", "MDT", "OST", and "SSD".

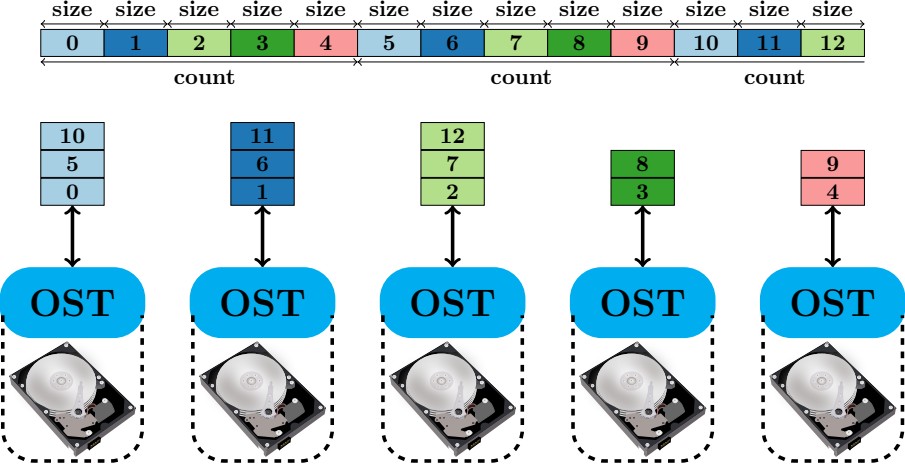

**Figure 5.** Schematic diagram of the striping of a file across multiple OSTs in a Lustre parallel file system. The "size" and "count" are the abbreviations of "stripe size" and "stripe count", respectively. In this example, the stripe count is five and the file is divided into 12 segments of a size equal to the stripe size. The number is the ID of a segment.

a finite number of OSTs. With a large stripe count, a file can be read from or written to multiple OSTs in parallel to achieve a high bandwidth and significantly improve the parallel IO performance.





### 3.2.2 Parallel IO Algorithm for Multiple Files

A restart file of the numerical model of a dynamical system contains the instantaneous states of the system and other auxiliary variables. In general, a DA system assimilates the available observations which only update some state variables but not all the variables in a restart file. Hence, it is desirable to update old restart files rather than to create new restart files from scratch. This way avoids copying the untouched variables from old restart files to new restart files and will further reduce the IO operations. As mentioned in Section 1, several high-level libraries for parallelly reading or writing a netCDF file are available currently, but only the flexible PIO (NCAR, 2018) supports update operations. One distinctive feature of the offline EnKF is that it needs to read $N_e$ restart files before computations and update these restart files after computations. These restart files have an identical structure. With this feature in mind, we propose an innovative algorithm to read and update multiple files with an identical structure. Figure 6 illustrates the parallel reading the state variables $\boldsymbol{x}_k$ from multiple restart files with an identical structure, the writing or updating is in the same manner except that scatter operations are changed to gather operations.

The algorithm for reading $N_e$ forecast ensemble files to the matrix $\mathbf{X}$, is such that, each member file is read into its corresponding column of the matrix $\mathbf{X}$. The rows of the matrix $\mathbf{X}$ are partitioned by $N_{mpi}$ MPI tasks. The information of this partition is passed from the DA module to the IO module as arguments so that the IO module and DA module have the same domain decomposition of the state vectors. The $N_{mpi}$ MPI tasks are partitioned by $N_{io}$ IO tasks in the IO module . For writing the matrix $\mathbf{X}$ to $N_e$ analyse ensemble files, the scatter operations are changed to gather operations.

There are two modes, the independent and collective mode, for all IO tasks to access a single shared file. With the independent mode, each IO task accesses the data directly from the file system without communicating or coordinating with the other IO tasks. This usually works best if the application is reading or writing large contiguous non-overlapping blocks of data in the file with one IO request because the parallel file systems do very well with an access pattern like that. In our proposed algorithm, an IO task reads or writes only one non-overlapping block of data in a file each time , so the independent IO mode is adopted.

Another advantage of this algorithm is that the MPI communication can be overlapped with the IO operation. For example in Figure 6, the IO task ① in a nonblocking way scatters the data read from the file 1 to the MPI tasks 1, 2, and 3; then shifts to read the file 2 without waiting for the previous scatter operation to finish. When the IO task ① finished its reading of the file 2, it checks, in most cases does not need to wait, the finish of the previous scatter operation since the MPI communication time is usually significantly shorter than the IO time; then in a nonblocking way scatters the data read from file 2 to the MPI tasks 1, 2, and 3; then shifts to read the next file in the same manner until all the files are read. Other IO tasks are in same manner. And the similar way is applied for the write or update operation. This almost eliminates the MPI communication time which significantly improves the performance of these parallel IO operations.

## 4 Experimental Environments, Designs, and Results

All the experiments are performed on the research supercomputer Beaufix in Météo France which is a Linux cluster built by BULL company. The SLURM system is used for the cluster management and the job scheduling. And this machine is equipped



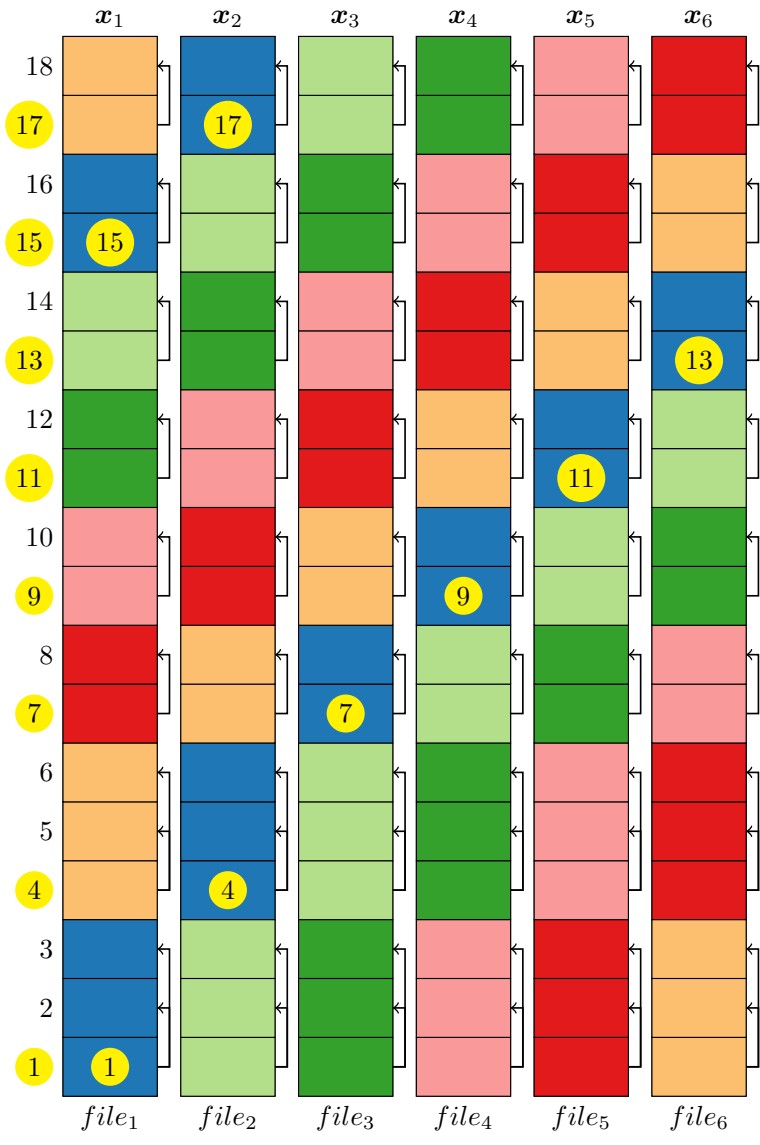

**Figure 6.** Schematic diagram illustrates the algorithm for reading $N_e$ ($N_e = 6$ for this example) forecast member files to a matrix $\mathbf{X} = [\boldsymbol{x}_1, \cdots, \boldsymbol{x}_6]$, that is, each member file is read into its corresponding column of the matrix $\mathbf{X}$. The numbers to the left of the first column are the ranks of the MPI tasks whose is in charge of the corresponding row of the matrix $\mathbf{X}$, and those with a yellow circle are the IO tasks. The cells with the same color are read simultaneously by the corresponding IO task, and then the IO tasks scatter the read-in data to the MPI tasks that they are charged of. In the first stage, the IO tasks of ①, ④, ⑦, ⑨, ⑪, ⑬, ⑮, and ⑰ read the cells with the purple color, and then, for example, the IO task of ① scatters the read-in data to itself and the MPI tasks of 2 and 3. In subsequent stages, each IO task performs a right circular shift by one column, then reads and scatters. Repeat this pattern until all the files are read.





with a highly scalable Lustre file system of 156 OSTs. The parallel IO algorithm developed by ourselves can use both PnetCDF or netCDF with parallel HDF5 as the backend. PnetCDF 1.10.0 is adopted for all the experiments in this study.

PDAF is an open-source parallel data assimilation framework which provides full implemented data assimilation algorithms, in particular ensemble-based Kalman filters like LETKF and LESTKF. PDAF is optimized for the large-scale applications run
on big supercomputers in both research and operational contexts. We chose PDAF as the basis to implement the proposed offline and online EnKFs using the efficient methods described in section 3 because it has the interfaces for both offline and online modes. With this unified basis, the study comprehensively assesses the efficiency of the offline and online EnKFs in terms of the time-to-solution, job queuing time, and IO time. We refer the readers to PDAF website (PDAF, 2018) for more detailed information.

## 4.1 Assessing the Proposed Parallel IO Algorithm

### 4.1.1 Experiments for Assessing the Proposed Parallel IO Algorithm

The key advantage of the Lustre file system is that it has many parameters which can be tuned by the user to maximize the IO performance according to the characteristics of the files and the configuration of the file system. The most relevant parameters are the stripe size and the stripe count. The Lustre manual provides some guidelines on how to tune these parameters. It is
interesting to see how the different combinations of the stripe size and the stripe count affect the performance of the proposed parallel IO algorithm with different numbers of IO tasks. Moreover, it is practical to determine the reasonable combination of these parameters by trial and error. Thus, a simple program using the proposed IO algorithm, which parallelly reads 40 files (each file is about 5 gigabytes in size) into a matrix $\mathbf{X}$ as illustrated in Figure 6 and then parallelly writes the matrix $\mathbf{X}$ back to the 40 files, is developed to record the IO times and the MPI communication times for each run. Each experiment is run with
1024 MPI tasks and takes a different combination of the stripe size, the stripe count, and the number of IO tasks. The stripe size can be 1, 2, 4, 8, 16, 32, and 64 megabytes. The stripe count can be 1, 2, 4, 8, 16, 32, 64, and 128. And the number of IO tasks can be 1, 2, 4, 8, 16, 32, 64, 128, 256, 512, and 1024. Thus, there are 616 experiments in total.

### 4.1.2 Performance of the Proposed Parallel IO Algorithm

Figure 7 shows how the combination of the stripe count and size has an influence on the IO performance. An obvious feature
in Figure 7 is that the IO times are always large when the stripe count is small regardless of the stripe size (e.g. when the stripe count is 1 or 2). This is reasonable because the small stripe count means small number of OSTs are used for storing the file, that is, it prevents high concurrent IO operations. But if the number of IO tasks for the file is significantly larger than the number of OSTs, the heavy competitions of IO tasks for the same OST actually increase the IO times substantially. On the other hand, the IO time with a small stripe size but a large stripe count gradually decreases as the increase of IO tasks (see the
cases with the stripe size of 1 or 2 megabytes but the stripe count of 64 or 128 in Figure 7). A small stripe size but a large stripe count means there are many small blocks of the large file (about 5 gigabytes in this case) distributed over many OSTs, that is, each IO task needs to perform IO operations over a large number of OSTs when the number of IO tasks is small; increasing





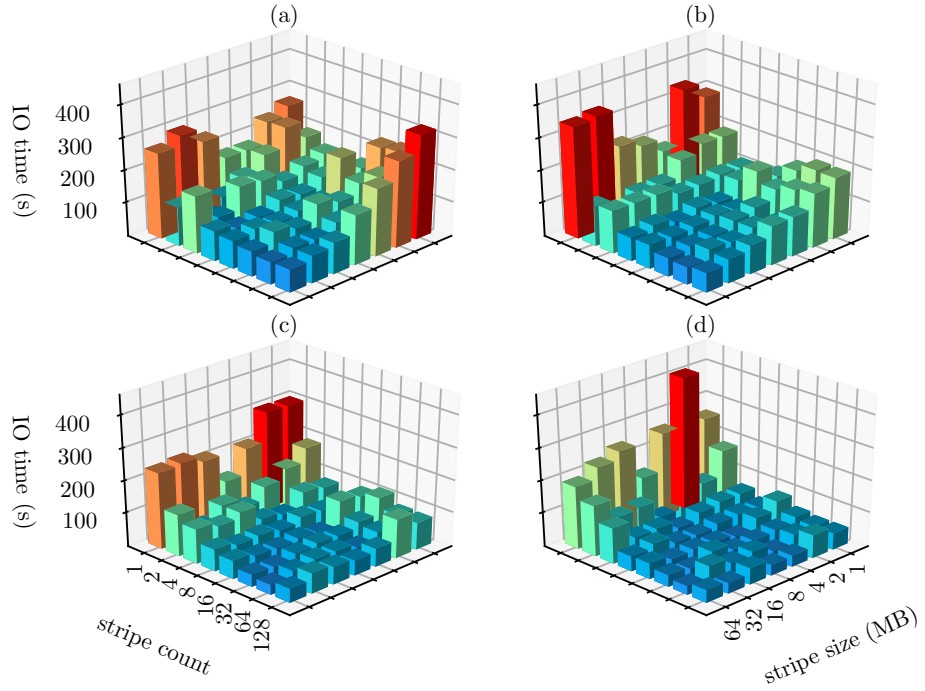

**Figure 7.** IO times of different combinations of the stripe count and stripe size with 32 (a), 64 (b), 128 (c), and 256 (d) IO tasks of 1024 MPI tasks for reading and writing 40 restart files using the proposed IO algorithm illustrated in Figure 6. The size of each restart file is about 5 gigabytes.

the number of IO tasks reduces the number of OSTs on which each IO task operates, thus reduces the IO times. For the same reason, a large stripe count allows high concurrent IO operations and fewer competitions, a large stripe size further reduces the number of OSTs on which one IO task operates when the file is large; therefore the combination of a large stripe count and a large stripe size with a large number of IO tasks generally reduces the IO time for a large file, as is evident in the four

subfigures of Figure 7 since all the IO times converge to the least with a stripe count of 128 and a stripe size of 64 megabytes. These imply the combination of the large stripe count with the large stripe size usually produces a small IO time for a large file. These suggest that it is important to have a consistent combination of the stripe count and the stripe size in line with the size of the file and number of IO tasks for a better IO performance.

  In Figure 7, the best IO performance is obtained with a stripe count of 128 and a stripe size of 64 megabytes for the cases of

10 32, 64, 128, and 256 IO tasks. For other cases of different numbers of IO tasks, a similar pattern is obtained (figures not shown). Owing to the smaller size of files, the stripe count of 128 and the stripe size of 1 megabytes are chosen as the combination of these two parameters with 40 (160) IO tasks for the medium (large) experiments described in Section 4.2.1 to compare the offline and online EnKFs.

  The IO throughput is the amount of data read or written per second. The upper panel of Figure 8 shown the IO time and

15 the IO throughput vary as a function of the number of IO tasks. The IO time and the IO throughput are the average of all the





616 experiments described in Section 4.1.1 and group by the number of IO tasks. The IO times (blue line in the upper panel of Figure 8) decrease quickly from about 1500 seconds to about 60 seconds as the increase of the number of IO tasks, then maintain nearly constant with a large number of IO tasks. The best IO performance is achieved with 1024 IO tasks; it takes about 60 seconds to read and write 40 restart files (each file has a size of about 5 gigabytes). For the same reason of the smaller size of files, the number of IO tasks is set to 40 (160) for the medium (large) scale experiments described in Section 4.2.1 to compare the offline and online EnKFs. And the variance of the IO times is large with a large number of IO tasks. The IO throughput increases gradually as the increase of the number of IO tasks. The maximum IO throughput is more than 1500 megabytes per second. Because the IO throughput is the average of different combinations of the stripe count and the stripe size, it can be beyond 2000 megabytes per second with the optimal combinations of these two stripe parameters (not shown). The variance of the IO throughput is proportional to the IO throughput. It is interesting to find that the proposed IO algorithm scales well since we do not see an apparently saturated IO time up to 1024 IO tasks.

In the lower panel of Figure 8, the IO time in the upper panel is decomposed into the time of opening and closing, the time of reading and writing, and the MPI communication time. The reading and writing time is dominant and its pattern is similar to that of the IO time in the upper panel. It is at least two orders of magnitude larger than the other two terms. The opening and closing time is slightly oscillating around 3 seconds. This opening and closing time is somewhat larger than that in local filesystem because the Lustre clients need to communicate with the metadata servers. The MPI communication time decreases as the increase of the number of IO tasks, and it is larger (smaller) than the opening and closing time with a small (large) number of IO tasks. Even though the MPI communication is overlapped with the IO operations, there is a waiting for the finish of the last reading or the first writing in our proposed algorithm. Thus, we believe that the major MPI communication time is dominated by this waiting time. Otherwise, the MPI communication time should be negligible if it is completely hidden behind the IO operations.

The impact of the stripe parameters on the IO performance depends on many factors such as the configuration and hardware of a Lustre system, the number and size of files to be read or written, and so on. So the exact value of the IO performance might vary with the situation of applications, but the statistics should have given some meaningful insights into how these parameters affect the IO performance and what is the optimal combination for this situation.

## 4.2 Comparing the Offline and Online EnKFs

### 4.2.1 Experiments for Comparing the Offline and Online EnKFs

The ultimate goal of this study is to develop an offline framework for high-dimensional ensemble Kalman filters which is at least as efficient as, if not faster than, its online counterpart in terms of the time-to-solution. Table 1 summarizes the experiments for comparing the offline and online EnKFs. The number of ensemble members is 40 for all experiments in Table 1. All the experiments use the same number of MPI tasks for each ensemble member regardless of the mode (offline or online) of the EnKF so that the model time and the analysis time are comparable. For example in Table 1, the medium and large scale problems use one node per member and four nodes per member, respectively. Since each node of our supercomputer has 40



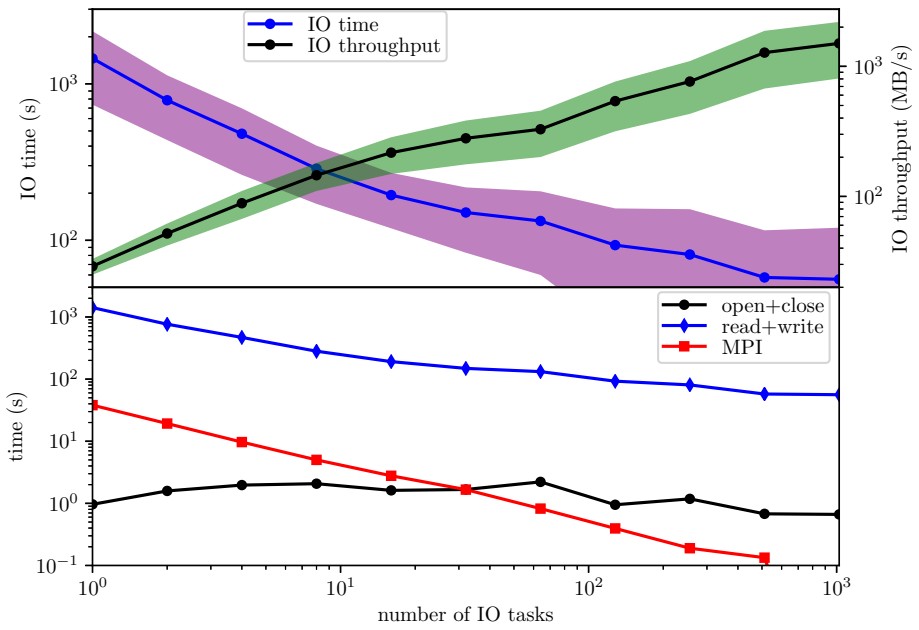

**Figure 8.** In the upper panel, the IO time (blue line) and throughput (black line) vary as a function of the number of IO tasks for reading and writing 40 restart files with 1024 MPI tasks using the proposed IO algorithm illustrated in Figure 6. The shadings indicate the ranges between plus and minus one standard deviation. The size of each restart file is about 5 gigabytes. In the lower panel, the IO time in the upper panel is decomposed into the time of opening and closing (black line), the time of reading and writing (blue line), and the MPI communication time (red line).

cores, the online EnKF requires 1600 (6400) MPI tasks for the medium (large) scale problem. But the number of MPI tasks for the offline EnKF dynamically ranges from 40 (160) to 800 (3200) for the medium (large) problem depending on the available nodes during its runnings. The large scale problem requires a large number of computer nodes which may imply a long queuing time for the simultaneous availableness of such a large number of nodes, but has a lower IO cost for the online mode. In contrast,

5   our proposed offline framework does not require all the computer nodes for all the members to be available simultaneously, but the IO cost may be high because of the intermediate outputs between the forecast phase and the analysis phase. Thus, a medium scale problem and a large scale problem are designed to address the dependence of time-to-solution on the scale of the problem. The medium and large scale problems are land grid points of a global field with a $0.1°$ and $0.05°$ resolution, respectively. The size of the state vector for the medium and large scale problems are 2127104 and 8498681, respectively. Each

10  of all the experiments in Table 1 is repeated 15 times, which are equivalent to 15 assimilation cycles, to obtain a robust statistics of measured times. As in real scenarios, other auxiliary variables besides the state variables, such as the location position and patch fraction, are needed to be read for the full functionalites of the model and DA. The corresponding restart files including the auxiliary variables are about 0.3 GB and 1.0 GB for the medium and large scale problems, respectively. Thus, both the offline and online EnKFs read and update all the variables in the restart files to assess their performances to a limit.



Both the background memebers and the observations are synthetic data in these experiments for both the offline and online EnKFs. These synthetic data are formed by the land grid points of the idealized global fields described in the following. The horizontal resolution of the global field is $\Delta_x = \frac{2\pi}{n_x}$ and $\Delta_y = \frac{\pi}{n_y}$ where the $n_x$ and $n_y$ are the number of grid points in longitude and latitude, respectively. The value of $n_x$ ($n_y$) is 3600 (1800) and 7200 (3600) for the medium and large scale

problems, respectively. The ensemble members and observations are generated from the following hypothetical true state (see Figure 9a):

$$state_{i,j}^t = \sin\left(-3 + \frac{4 \cdot i \cdot \Delta_x}{2\pi - \Delta_x}\right)^2 \cos\left(-2 + \frac{4 \cdot j \cdot \Delta_y}{\pi - \Delta_y}\right)^3 \tag{10}$$

The members (figures not shown) are generated by randomly shifting the true state in longitude:

$$state_{i,j}^k = \sin\left(-3 + \frac{4 \cdot i \cdot \Delta_x}{2\pi - \Delta_x} + \Delta_s\right)^2 \cos\left(-2 + \frac{4 \cdot j \cdot \Delta_y}{\pi - \Delta_y}\right)^3 \tag{11}$$

where the superscript $k \in [1, N_e]$ denotes the ID of a member, $i \in [0, n_x - 1]$ and $j \in [0, n_y - 1]$ are the longitude and latitude index of the grid point, respectively, and $\Delta_s$ is a shift drawn from a uniform distribution on $[-0.5, 0.5]$. The observations (see Figure 9d) are the true state values plus the observation errors at the grid points randomly picked from the total grid points. The number of observations is equal to $10\%$ of the number of the total grid points, and the observation errors are drawn from a normal distribution with a mean of zero and a variance of $0.25^2$. Thus, the observation operator simply becomes $\mathcal{H}(x) \equiv x$.

All these fields are written to the corresponding NetCDF files in advance so that the offline or online EnKFs can read them at the beginning of each cycle.

All the assimilation experiments use the LESTKF scheme with a localization radius of $50°$, and the localization scheme of Nerger et al. (2012b). For the sake of experiments, the model simply reads its initial condition, sleeps one second, and writes its restart file for the offline mode or sends its states to the DA component for the online mode. In the offline mode, each model

member reads its corresponding initial condition and writes the corresponding restart file. Then the DA component reads the restart files, performs the analysis, and writes the analysis ensemble files. In the online mode, each model member only reads its corresponding initial conditon, and the DA component writes the analysis ensemble files. All these IO operations are done by the proposed parallel IO algorithm which certainly can read or write one file or multiple files in parallel. This make it possible to fairly compare their IO times. The jobs of the first assimilation cycle of the offline and online EnKFs for the medium scale

problem are submitted at the same time, the jobs of next cycle are submitted without any delays after the completion of the previous cycle, this repeats until the last cycle; so does the large scale problem. This manner guarantees the fair comparison of the queuing times since the offline and online EnKFs are in the same loaded conditon of the supercomputer.

### 4.2.2   Results of Comparing the Offline and Online EnKFs

Figure 9b is the analysis mean $\overline{x^a}$ obtained by the offline or online EnKF. Compared to the initial state (Figure 9c) which is

the ensemble mean $\overline{x^f}$ before the assimilation, it can be seen that the analysis mean $\overline{x^a}$ (Figure 9b) is significantly close to the true state (Figure 9a), especially over the northern Canada, Greenland, and northwestern Africa. The only difference between the offline and online EnKFs is the coupling mode which only affects the time-to-solution, so they produce identical analysis





**Table 1.** Experiments for Comparing the Offline and Online EnKFs

|  | **Medium problem** | **Large problem** |
|---|---|---|
|  | state vector size=2127104 | state vector size=8498681 |
|  | job time limit=20 minutes | job time limit=80 minutes |
|  | one restart file size=0.3 GB | one restart file size=1.0 GB |
| **Offline** | 20 jobs and 1 node/job | 20 jobs and 4 nodes/job |
| **Online** | 1 job and 40 nodes/job | 1 job and 160 nodes/job |

results. Therefore, the following evaluations focus on the differences of the times between the offline and online EnKFs. In a research context, the queuing time is largely dependent on the loaded condition of the supercomputer, so the time-to-solutions of all the experiments in Table 1 are assessed both in the time such as during the weekend when the supercomputer is low-loaded and in the time such as during the weekday when the supercomputer is high-loaded.

In the offline mode for the assimilation cycle $j$, each model member whose ID is $k$ records its running time $t_{jk}^m$, the actual executing time, which includes its IO time $t_{jk,IO}^m$; and the DA component records its running time $t_j^a$ which includes its IO time $t_{k,IO}^a$. Thus, the running time and the IO time of the assimilation cycle $j$ are

$$t_{j,running} = \frac{\sum_{k=1}^{k=N_e} t_{jk}^m}{N_e} + t_j^a \tag{12}$$

and

$$t_{j,IO} = \frac{\sum_{k=1}^{k=N_e} t_{jk,IO}^m}{N_e} + t_{j,IO}^a, \tag{13}$$

respectively. In the online mode, $t_{j,running}$ and $t_{j,IO}$ are explicitly recorded by the online EnKF owing to the online coupling of the model and the DA component.

    Thus, the average running time and the average IO time of an assimilation cycle are calculated as

$$\bar{t}_{running} = \frac{\sum_{j=1}^{j=15} t_{j,running}}{15} \tag{14}$$

and

$$\bar{t}_{IO} = \frac{\sum_{j=1}^{j=15} t_{j,IO}}{15}, \tag{15}$$

respectively. Similarly, the average queuing time of an assimilation cycle is

$$\bar{t}_{queuing} = \frac{\sum_{j=1}^{j=15} t_{j,queuing}}{15} \tag{16}$$

where $t_{j,queuing}$ is the queuing time of the first running job in the assimilation cycle $j$. Since this study is interested in the

time-to-solution, the EnKF system records the elapsed time from the beginning to the end of 15 assimilation cycles as the

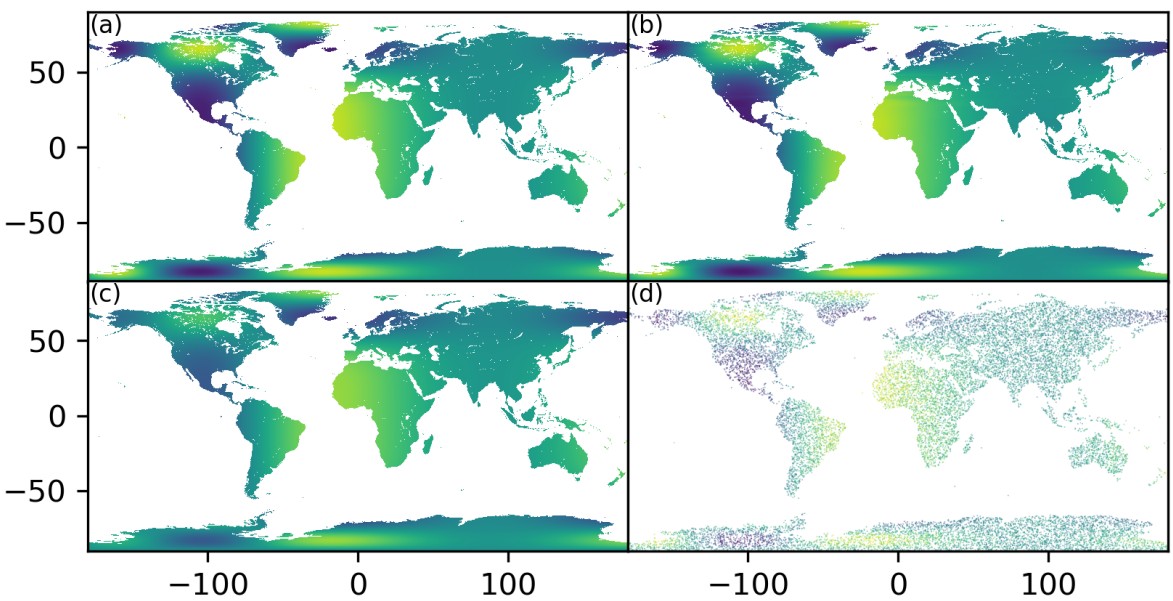

**Figure 9.** The synthetic fields of the true state (a), the posterior state (b) after the assimilation, the prior state (c) before the assimilation, and the observations (d) for the medium scale problem. Please refer to the second paragraph of Section 4.2.1 for the generations of these synthetic fields.

time-to-solution $t_{solution}$. Thus, the average of the total time of an assimilation cycle is $\bar{t}_{total} = \frac{t_{solution}}{15}$. Except for the total time, the standard deviation can be calculated as

$$\sigma_x = \sqrt{\frac{\sum_{j=1}^{j=15}(t_{j,x} - \bar{t}_x)^2}{15}} \qquad (17)$$

where $x$ can be "queuing", "running", or "IO". Thus, $\bar{t}_{total}, \bar{t}_{queuing}, \bar{t}_{running}$, and $\bar{t}_{IO}$ correspond to the columns of "total",

"queuing", "running", and "IO" in Table 2, respectively. Table 2 summarizes these average times of the offline and online EnKFs for both medium and large scale problems in both low and high-loaded situations. Figure 10 shows the statistics of the total time, the queuing time, and the running time of 15 assimilation cycles for both medium and large scale problems in both low-loaded and high-loaded conditions. It can be seen from Figure 10 that the running time of the offline EnKF is the same as that of the online EnKF for both medium and large scale problems regardless of the loaded conditions of the supercomputer.

In the low-loaded condition (Figure 10a and b), it is surprising that the IO time of the offline EnKF is about 29% (37%) larger than that of the online EnKF for the medium (large) problem. In principle, the former should be twice as large as the later. The possible explanation is that this IO time might be affected by the jitter of the supercomputer including the underlying networks and the Lustre file system. From Table 2, it can be shown that the IO time of the offline (online) EnKF only accounts for a fraction of about 6.6% (2.8%) and 2.4% (1.3%) of the total time for the medium and large scale problems, respectively. It

is obvious that with the proposed IO algorithm, the IO time becomes a less severe problem as the scale of the problem increases





since the analysis time becomes dominant. The queuing time is slightly less than the running time for the offline EnKF, but the queuing time is two to four times larger than the running time for the online EnKF. Even in such a low-loaded condition, it is evident that the offline mode has a shorter queuing time than the online mode because the online mode simultaneously requires significantly more nodes than the offline mode. Thus, the offline EnKF is faster than the online EnKF in terms of the

time-to-solution. In the limit of zero queuing time, the former is at least as fast as the later. Therefore, the dynamically running job scheme described in Section 3.1 does reduce the queuing time. In other words, it can be shown from Table 2 that the offline mode is nearly 45% (26%) faster than the online mode in terms of time-to-solution for the medium (large) scale problem.

In the high-loaded condition (Figure 10c and d), the IO times increase a bit owing to the loaded condition of the underlying Lustre file system. Even the IO time for the medium scale problem is larger than that for the large scale problem, it implies

that loaded conditions also affect IO performances. But it can be shown from Table 2, the IO time of the offline (online) EnKF is still as small as a fraction of about 8.8% (2.8%) and 1.4% (0.4%) of the total time for the medium and large scale problems, respectively. Except for the offline mode for the medium scale problem, the queuing times (especially for the online mode) are substantially larger than the running time. For the medium problem (Figure 10c), the queuing time of the offline mode is even less than the running time because it is common that there are some dispersed nodes available in a high-loaded supercomputer.

The offline mode which requires few nodes can quickly obtain the available nodes to start its running. For the large problem (Figure 10d), the queuing time of the offline mode is a factor of around 1.7 (estimated from Table 2) larger than the running time. On the contrary, the queuing time of the online mode is a factor of around 6.0 and 10.1 (estimated from Table 2) larger than the running time for the medium and large problems, respectively. In such an occasion, the queuing time dominates the time-to-solution, thus the offline mode is significantly faster than the online mode. Thus estimated from Table 2, the offline

mode is about 55% and 67% faster than the online mode in the high-loaded condition.

Comparing the queuing times for the large scale problem in Table 2, it can be seen that in a high-loaded condition they are several times larger than those in a low-loaded condition. The queuing time becomes dominant for a large scale problem in a high-loaded supercomputer. The offline EnKF is significantly faster than the online EnKF in terms of time-to-solution. As the numerical model is getting a higher and higher resolution, the offline EnKF might be a better option than the online EnKF for

a high-dimensional system in terms of time-to-solution, at least in a research context.

From Figure 10, it can be seen that the variances of both the running time and the IO time are negligible, but the variance of the queuing time is even larger than its average value except for the large scale problem in the high-loaded condition. This means the instantaneous loaded conditon of the supercomputer varies greatly even in the low-loaded condition. A careful examination of the recorded times highlights that the large variance is come from the extremely large queuing time of one or

two cycles. Because of this high varied loaded condition, the dynamically running job scheme has its place to play its strength.

To summarize, the offline mode is faster than the online mode in terms of time-to-solution for an intermittent data assimilation system because the queuing time is dominant and the IO time only accounts for a small fraction of the total time with the proposed IO algorithm. Even in the situation where the queuing time is negligible, the offline mode can be at least as fast as the online mode with the proposed IO algorithm and the dynamically running job scheme. The queuing times as well the



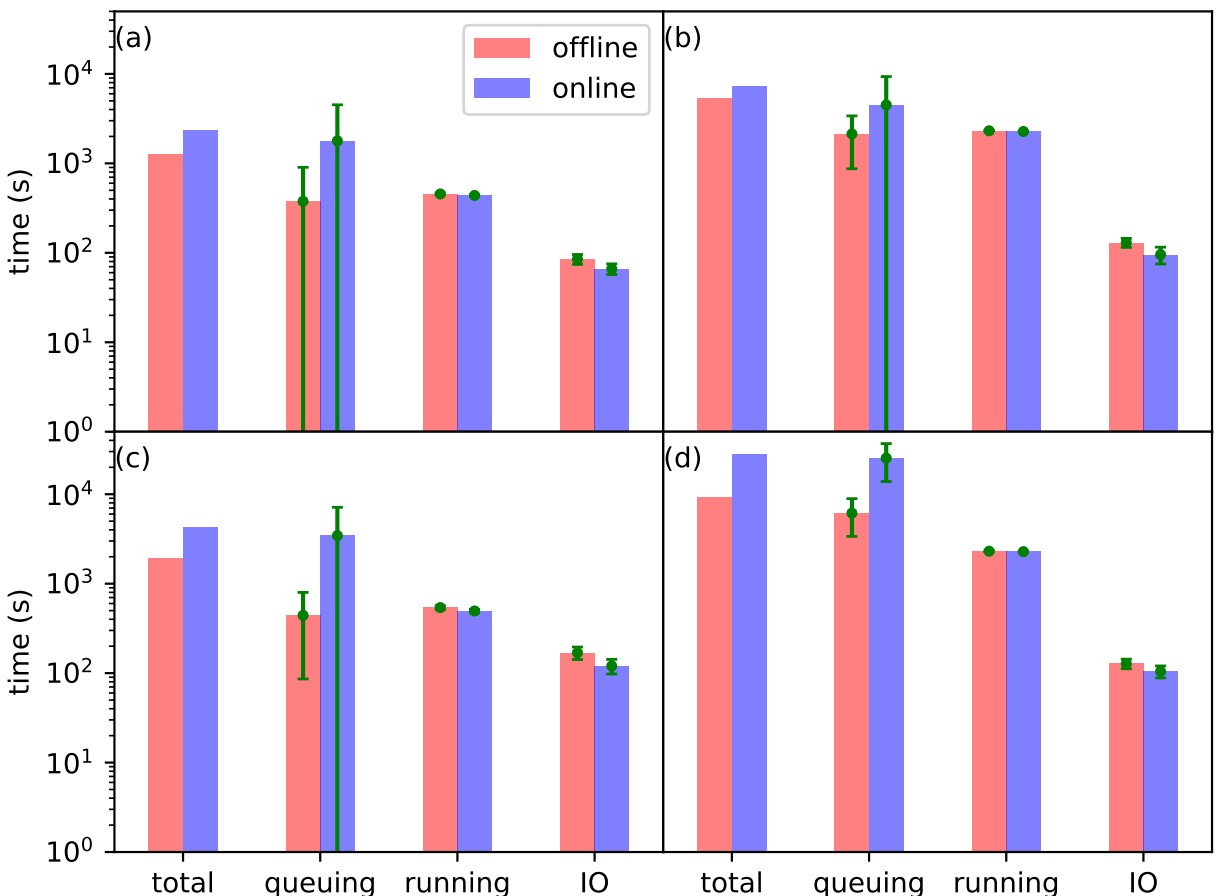

**Figure 10.** The average total time, queuing time, running time, and IO time of the offline (red bars) and online (blue bars) EnKFs for the low-loaded (a and b) and high-loaded (c and d) conditions. The panels (a and c) and (b and d) are for the medium and large scale problems, respectively. The green line indicates the corresponding standard deviation.

total times vary as the loaded conditions of a supercomputer, but these statistics shed some insights on how the queuing time influences the time-to-solution of an EnKF system.

## 5 Conclusion and Discussion

With the sophisticated dynamically running job scheme and the innovative parallel IO algorithm proposed in the study, a comprehensive assessment of the total time, the queuing time, the running time, and the IO time between the offline and online EnKFs for medium and large scale assimilation problems is presented for the first time. This study not only provides the detailed technical aspects for an efficient implementation of an offline EnKF but also presents the thorough comparisons





**Table 2.** The average times in seconds of the offline and online EnKFs in low and high-loaded situations for medium and large problems. The number in the parentheses is the percent to the corresponding total time.

|  |  | Medium problem | | | | Large problem | | | |
|---|---|---|---|---|---|---|---|---|---|
|  |  | total | queuing | running | IO | total | queuing | running | IO |
| low-loaded | offline | 1286 | 377 (29%) | 455 (35%) | 85 (6.6%) | 5381 | 2132 (40%) | 2314 (43%) | 130 (2.4%) |
|  | online | 2352 | 1786 (76%) | 438 (19%) | 66 (2.8%) | 7301 | 4500 (62%) | 2271 (31%) | 95 (1.3%) |
| high-loaded | offline | 1920 | 442 (23%) | 540 (28%) | 169 (8.8%) | 9303 | 6150 (66%) | 2306 (25%) | 128 (1.4%) |
|  | online | 4266 | 3447 (81%) | 495 (12%) | 120 (2.8%) | 28110 | 25337 (90%) | 2282 ( 8%) | 104 (0.4%) |

between the offline and online EnKFs in terms of time-to-solution which opens new possibilities to re-examine the applicable conditions of the offline and online EnKFs. The main conclusions from the experimental results are as follows:

1. The proposed parallel IO algorithm can drastically reduce the IO time for reading or writing multiple files with an identical structure. The tuning parameters of a stripe count and a stripe size should be consistent, and high values of these two parameters usually allow high concurrent IO operations and low competitions which significantly reduce the IO time.

2. The running times of both offline and online EnKFs for high-dimensional problems are almost the same since the IO time only accounts for a small fraction which further decreases as the increase of the scale of the problem. This implies that the proposed parallel IO algorithm is very scalable.

3. In a low-loaded supercomputer, the queuing time might be equal to or less than the running time, but the offline EnKF is at least as fast as, if not faster than, the online EnKF in terms of the time-to-solution because the offline mode requires less simultaneously available nodes and more easily and quickly obtains the requested nodes to reduce the queuing time than the online mode.

4. In a high-loaded supercomputer, the queuing time is usually several times larger than the running time, thus the offline EnKF is substantially faster than the online EnKF in terms of time-to-solution because the queuing time is dominant in such a circumstance.

5. The loaded condition of a supercomputer varies greatly which justifies the dynamically running job scheme of an offline EnKF.

It is evident that the offline EnKF can be as fast as, if not faster than, the online EnKF. On average, the offline mode is significantly faster than the online mode in the research context. Even in the operational context where the queuing time can be negligible, the offline mode still has an advantage over the online mode. This is because the online mode never have a chance to run when the total nodes required are larger than the total nodes of a supercomputer if the number of members is so large.



In general, the observations are only available at a regular time interval, that is, not every time step of the numerical model has observations for the assimilation. Thus, most DA systems are an intermittent system. Therefore, with a good implementation and a high performance parallel file system, an offline mode is still preferred with the perspective of the techniques proposed in this study because of their easy implementations and promising efficiencies. In climate modelling context, even the assimilation

is intermittent, an online mode might be appropriate because the model can run a very long time once it has started. The running time substantially outweighs the queuing time.

In terms of job managements, other job schedualling systems are similar to the one (SLURM) used in this paper, so the dynamically running job scheme also works for these systems and can be adapted with minor changes. Other parallel file systems may be different from the Lustre parallel file system in many aspects. But in principle they all have a feature to

distribute a file over multiple storage devices for supporting concurrent IO operations. And the proposed parallel IO algorithm does not rely on any specific characteristics of the Lustre parallel file system, that is, similar conclusions could be obtained for other parallel IO file system. Thus, we believe that the techniques proposed in this paper can be generalized to other supercompers, even to the future supercomputer architectures.

For a high-dimensional system with a large number of ensemble members, the total size of the output files is extremely big.

This poses a great burden to archive these files. Even though the archiving is not a critical component of an EnKF system, the time-to-solution can be further reduced if the archiving is implemented properly. We also implemented a very practical method to asynchronously archive the output files to a massive backup server with compressing and transferring on the fly. This method further reduce the time-to-solution of an EnKF system. The details of this method are beyond the scope of this paper. The techniques proposed in this paper are being incorporated into the offline framework of LDAS-Monde at Météo

France.

*Code availability.*   PDAF is publicly available at http://pdaf.awi.de. The offline and online EnKFs built on the top of PDAF for all experiments presented in this paper are available at https://doi.org/10.5281/zenodo.2703420.

*Competing interests.*   The authors declared no competing interests.

*Author contributions.*   Y. Zheng designed and implemented the dynamically running job scheme and the parallel IO algorithm with discus-

sions from C. Albergel, S. Munier, and B. Bonan. Y. Zheng implemented the offline and online EnKF systems. Y. Zheng designed and carried out the experiments. Y. Zheng prepared the paper with contributions from all coauthors.





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
