# Peer review of "An Offline Framework for High-dimensional Ensemble Kalman Filters to Reduce the Time-to-solution"

_Geoscientific Model Development, 2019_

## Referee Comment (RC1) · Anonymous Referee #1 · 30 Jul 2019

The paper "An offline Framework for High-dimensional Ensemble Kalman Filters to Teduce the Time-to-solution" by Zheng et al. is, as far as the reviwer is aware, the first empirical study of the maximized wall-clock efficiency of both online and offline approaches to ensemble Kalman filterting as used in an operational context.

**1   General Comments**

- The paper utilizes the LESTKF, yet the only mathematical description is of a general ETKF-like filter. A subsection on how the ESTKF, and a section on localiza-

tion (and implementation challanges with localization therein) would greatly help the reader.

- In the experimental design section (4.2.1 in this draft), the choice of randomly selecting the observation points is concerning, a more uniform, or atleast reproducible approach would instill more confidence in the methodology.

- Again in the experimental design, if the aim is to reproduce operational conditions, why is more realistic data, say generated from some model like SPEEDY or WRF not used?

- Again in the experimental design section, operationally we consider the amount of observations as being three orders of magnitude lower than the state space, yet the choice made in this paper is only one order of magnitude lower. This might bias the results in favor of the offline approach.

- Equation 17 on page 22 should have 14 in the denominator as the mean is estimated, and not exactly known. This fact is used earlier in the EnKF description.

- In the conclusions (section 5 in this draft) maybe don't use bullet points, and try to more fluidly outline the main results? Though this is not that much of an issue.

**2 Technical Corrections**

- p1l17, 'for intermittent'.

- p3l32, 'demands substantial'

- Figure 7 is of a particularly low DPI, and looks jarring compared to the other figures. Perhaps a flat 2D figure could convey the same information more clearly?

• p22l11, maybe use 'longer' instead of 'larger' in reference to time?

---

## Author Comment (AC1) · 5 Dec 2019

**An Offline Framework for High-dimensional Ensemble Kalman Filters to Reduce the Time-to-solution**

Yongjun Zheng[1], Clément Albergel[1], Simon Munier[1], Bertrand Bonan[1], and Jean-Christophe Calvet[1]

[1]CNRM, Université de Toulouse, Météo-France, CNRS, Toulouse, France

December 5, 2019

Dear reviewer,

Thank you very much for your comments. The followings in blue are our responses. The texts in red are those we revised or added in our new manuscript.

Best regards,

Yongjun ZHENG on behalf of the authors
* * *
Reviewer #1:

The paper "An offline Framework for High-dimensional Ensemble Kalman Filters to Teduce the Time-to-solution" by Zheng et al. is, as far as the reviwer is aware, the first empirical study of the maximized wall-clock efficiency of both online and offline approaches to ensemble Kalman filterting as used in an operational context.

Thank you for your time and effort to read and comment our paper carefully.

**1  General Comments**

The paper utilizes the LESTKF, yet the only mathematical description is of a general ETKF-like filter. A subsection on how the ESTKF, and a section on localization (and implementation challanges with localization therein) would greatly help the reader.

The main purpose of this study is to compare the time-to-solution of offline and online ensemble Kalman filters. Even though the results presented in this paper are obtained by using the LESTKF, the conclusions can be generalized to other EnKF schemes as well. For example, the conclusions are still held when using LETKF scheme (results are not shown in the paper). The LESTKF is adopted in this paper because the PDAF documents recommend it and we thought the LESTKF may be well tested in PDAF. The following sentence is added near line 18 of page 20 in the new manuscript for referring readers to those papers on the full description (Neger et al, 2012a) of the ESTKF and its localizations (Nerger et al, 2006): We refer readers to the

[Figure]

Figure 1: The average total time, queuing time, running time, and IO time of the offline (red bars) and online (blue bars) EnKFs, The panel (a) and (b) are for the medium and large scale problems, respectively. The green line indicates the corresponding standard deviation.

paper of Nerger et al. (2012a) for a full description of the ESTKF, and the paper of Nerger et al. (2006) for the domain and observation localizations using in LESTKF. This might be a proper way for the readers who are interested in how the ESTKF and its localizations work but also for the paper being succinctly around its subject.

In the experimental design section (4.2.1 in this draft), the choice of randomly selecting the observation points is concerning, a more uniform, or atleast reproducible approach would instill more confidence in the methodology.

Thank you very much for pointing out the reproducibility. Following your suggestion, we did an experiment by having one observation of every ten grid points regularly instead of selecting 10% of the grid points randomly, the results show no big differences in both the analysis performance and the time-to-solution (see Figure 1). Actually, we believe the randomly selecting method does give us a higher confidence since the observations are usually irregularly scattered over the domain.

Again in the experimental design, if the aim is to reproduce operational conditions, why is more realistic data, say generated from some model like SPEEDY or WRF not used?

This is a really good question. In fact, the initial purpose of this study is to test the possibility

of substituting an online EnKF for an offline one. First, building an online EnKF demands more substantial time and effort than building an offline one, we did not know whether it is worth investigating time to do so. Second, an online EnKF does reduce the IO time, but it was not clear whether an online EnKF is faster than an offline EnKF in terms of time-to-solution. Therefore, it is more reasonable to address these questions with a very simple and idealized model before seriously investigating time to build a new system. This is the main reason this study did not use a complex model.

Again in the experimental design section, operationally we consider the amount of observations as being three orders of magnitude lower than the state space, yet the choice made in this paper is only one order of magnitude lower. This might bias the results in favor of the offline approach.

We agree with the reviewer that the observations used in our paper are denser than those in nowaday operational practice. The large amount of observations certainly increases the running time of EnKF analysis, but the IO time is barely affected. As shown by Figure 10 in the paper, the dominant time is the queuing time, especially for a large scale problem. The queuing time can be stable in a fixed loaded condition but the running time decreases if less observations are used, that is, the ratio of queuing time to running time will become biger . This evidently further convinces that an offline EnKF wins the favor over its online counterpart.

Equation 17 on page 22 should have 14 in the denominator as the mean is estimated, and not exactly known. This fact is used earlier in the EnKF description.

Thank you very much for pointing out this typo. It has been corrected as $\sigma_x = \sqrt{\frac{\sum_{j=1}^{j=15}(t_{j,x}-\bar{t}_x)^2}{14}}$ in the new manuscript.

In the conclusions (section 5 in this draft) maybe don't use bullet points, and try to more fluidly outline the main results? Though this is not that much of an issue.

Thank you for the suggestion. This paragraph has been re-organized slightly without using bullet points in page 25 of the new manuscript as follows:
    In summary, the proposed parallel IO algorithm can drastically reduce the IO time for reading or writing multiple files with an identical structure. The tuning parameters of a stripe count and a stripe size should be consistent, and high values of these two parameters usually allow high concurrent IO operations and low competitions which significantly reduce the IO time. Using the proposed parallel IO algorithm, the running times of both offline and online EnKFs for high-dimensional problems are almost the same since the IO time only accounts for a small fraction which further decreases as the increase of the scale of the problem. This implies that the proposed parallel IO algorithm is very scalable. On the contrary, in a low-loaded supercomputer, the queuing time might be equal to or less than the running time, thus the offline EnKF is at least as fast as, if not faster than, the online EnKF in terms of the time-to-solution because the offline mode requires less simultaneously available nodes and more easily and quickly obtains the requested nodes to reduce the queuing time than the online mode. But in a high-loaded supercomputer, the queuing time is usually several times larger than the

running time, thus the offline EnKF is substantially faster than the online EnKF in terms of time-to-solution because the queuing time is dominant in such a circumstance. Therefore, The loaded condition of a supercomputer varies greatly which justifies the dynamically running job scheme of an offline EnKF.

**2    Technical Corrections**

- p1l17, 'for intermittent'.
- p3l32, 'demands substantial'
- Figure 7 is of a particularly low DPI, and looks jarring compared to the other figures. Perhaps a flat 2D figure could convey the same information more clearly?
- p22l11, maybe use 'longer' instead of 'larger' in reference to time?

Thank you very much for the very careful reading. In the new manuscript, all are corrected ("for the intermittent data assimilation" $\longrightarrow$ "for intermittent data assimilation", "demands a substantial time and effort" $\longrightarrow$ "demands a substantial time and effort", and "larger" $\longrightarrow$ "longer" ) as suggested except for the Figure 7. Figure 7 seems look good in our side. But we will try to improve it if the quality is not enough for publication.

---

## Referee Comment (RC2) · Anonymous Referee #2 · 10 Feb 2020

This work seeks to reduce the time to solution of an offline EnKF via a dynamically running job scheme and a parallel IO algorithm. Numerical results show that the offline EnKF is significantly faster than the online EnKF in terms of time-to-solution.

This reviewer finds the paper to be well written, and the ideas proposed to be novel. Therefore I recommend acceptance after minor revisions.

Minor comments:

The existing work on parallelizing EnKF is not well represented. For a more complete view of where the field is and where the current work stands, the reviewer suggests that the authors consider at least the following references, given here by their DOI:

[Figure]

10.1175/MWR-D-13-00011.1     10.1145/3293883.3295722     10.1137/16M1097031
10.1016/j.jocs.2017.04.005 10.1007/s10586-017-1407-1 10.1007/s10236-015-0888-9
10.5281/zenodo.1086985 10.1175/JTECH2049.1 10.1016/j.cageo.2012.10.007

---

## Editor Comment (EC1) · Adrian Sandu (Editor) · 10 Feb 2020

Please respond carefully to both reviewers' comments.

———————————————————

---

## Author Comment (AC2) · 20 Feb 2020

**An Offline Framework for High-dimensional Ensemble Kalman Filters to Reduce the Time-to-solution**

Yongjun Zheng[1], Clément Albergel[1], Simon Munier[1], Bertrand Bonan[1], and Jean-Christophe Calvet[1]

[1]CNRM, Université de Toulouse, Météo-France, CNRS, Toulouse, France

February 20, 2020

Dear reviewer,

Thank you very much for your comments. We have modified the manuscript according to your suggestions and comments, please see the attached new manuscript.

Best regards,

Yongjun ZHENG on behalf of the authors
* * *
This work seeks to reduce the time to solution of an offline EnKF via a dynamically running job scheme and a parallel IO algorithm. Numerical results show that the offline EnKF is significantly faster than the online EnKF in terms of time-to-solution. This reviewer finds the paper to be well written, and the ideas proposed to be novel. Therefore I recommend acceptance after minor revisions.

Minor comments:

The existing work on parallelizing EnKF is not well represented. For a more complete view of where the field is and where the current work stands, the reviewer suggests that the authors consider at least the following references, given here by their DOI:

10.1175/MWR-D-13-00011.1

10.1145/3293883.3295722

10.1137/16M1097031

10.1016/j.jocs.2017.04.005

10.1007/s10586-017-1407-1

10.1007/s10236-015-0888-9

10.5281/zenodo.1086985

10.1175/JTECH2049.1

10.1016/j.cageo.2012.10.007

Thank you very much for the comment and suggestion. All these papers are cited properly in the new manuscript (see L24-27 P2, L31-34 P2, L3-4 P3, and L23-24 P4). Also the new cited papers are shown in blue in the references in the new manuscript.

---

## Author Comment (AC3) · 20 Feb 2020

Dear Editor,

We have now replied to comments from two anonymous referees. We thank the two referees for their positive comments about our work and for their detailed reviews that has helped us to improve the quality of our manuscript. Attached are our answers in one single file.

Clément Albergel on behalf of the co-authors

[Figure]

Please also note the supplement to this comment:
https://www.geosci-model-dev-discuss.net/gmd-2019-132/gmd-2019-132-AC3-supplement.pdf

————————————————————

---

## Referee Report (RR1)

**Reviewer's Comments for gmd-2019-132**

After reading the paper, and clearly having in mind your research questions: Is an online EnKF really faster than an offline EnKF? Can an offline EnKF be as fast as, if not faster than, an online EnKF with a good framework and algorithms using advanced techniques of parallel IO? I think the paper can be improved in the following directions:

1. In section 1 (Introduction), you discuss efficient EnKF formulations: EnKF methods that exploit the rank-deficiency of ensemble covariances to come up with efficient EnKF formulations (i.e., by using Sherman Morrison, SVD, etc.), and efficient EnKF formulations which account for localization. You can enrich your paper by distinguishing between these two classes of filter derivations; it is not a good idea two present both families as a single one.

2. In section 2, I do not agree with the statement: *the Cholesky decomposition is more efficient than the SVD, but the SVD is more robust if the matrix is significantly ill-conditioned.* For unlocalized EnKF formulations, via the SVD decomposition, we can obtain EnKF implementations whose computational effort reads $O(n \cdot N^2)$, where $n$ is the number of model components and $N$ is the ensemble size. On the other hand, by employing a direct solution (i.e., by employing a Cholesky decomposition), we can get EnKF formulation whose analysis steps can be computed with $O(n^3)$ long computations.

3. As you may know, **localization methods** are crucial to getting accurate analysis states. These methods mitigate the impact of spurious correlations in EnKF based formulations. In operational contexts, when domain decomposition is employed during assimilation steps, the dimension of local boxes can still be much larger than that of ensemble sizes. Therefore, local analysis increments can be poorly estimated at each sub-domain. How does your proposed method deal with this? I would like you to increase the discussion in this direction, you have already cited two papers that deal with this:

    (a) Anderson, J. L. (2001). An ensemble adjustment Kalman filter for data assimilation. Monthly weather review, 129(12), 2884-2903.

(b) Nino-Ruiz, E. D., Sandu, A., & Deng, X. (2019). A parallel implementation of the ensemble Kalman filter based on modified Cholesky decomposition. Journal of Computational Science, 36, 100654.

4. While localization methods reduce the impact of spurious correlations, **covariance inflation** mitigates the impact of the under-estimation of sample variances. Most of the square-root based formulations (i.e., ETKF, and LETKF) suffer from under-estimation of sample variances, but, you do not mention this relevant topic in your entire paper.

5. When domain decomposition is employed, it is very common to send boundary information to obtain physically consistent solutions, in this manner, solutions at different processors do not look like independent domains but, they are "*connected*" since neighboring sub-domains share boundary information. I may misunderstand but, are you sharing boundary information among neighboring sub-domains? Thus far, it seems like not, right? If this is correct, how do you guaranty that global solutions (once all local analysis are mapped back onto the global domain) are consistent with the physics and the dynamics of numerical models?.

6. What observational network do you employ during experiments? are employing a full observational network (all model components are observed)? What is the performance of your method (in terms of accuracy) as the observational network becomes sparse?

 I hope these comments help to improve your paper.

---

## Author Response (AR2)

**An Offline Framework for High-dimensional Ensemble Kalman Filters to Reduce the Time-to-solution**

Yongjun Zheng[1], Clément Albergel[1], Simon Munier[1], Bertrand Bonan[1], and Jean-Christophe Calvet[1]

[1]CNRM, Université de Toulouse, Météo-France, CNRS, Toulouse, France

June 19, 2020

Dear reviewer,

We really appreciated your comments and suggestions to help us enrich and improve our paper. The followings are our responses in blue, together with the modified manuscript with highlighting modifications.

Best regards,

Yongjun ZHENG on behalf of the authors
* * *
After reading the paper, and clearly having in mind your research questions: Is an online EnKF really faster than an offline EnKF? Can an offline EnKF be as fast as, if not faster than, an online EnKF with a good framework and algorithms using advanced techniques of parallel IO? I think the paper can be improved in the following directions:

Thank you for the positive comment.

1. In section 1 (Introduction), you discuss efficient EnKF formulations: EnKF methods that exploit the rank-deficiency of ensemble covariances to come up with efficient EnKF formulations (i.e., by using Sherman Morrison, SVD, etc.), and efficient EnKF formulations which account for localization. You can enrich your paper by distinguishing between these two classes of filter derivations; it is not a good idea two present both families as a single one.

   Thanks for the suggestion. This study focuses on the comparison of the computational efficiencies between offline and online EnKFs. As long as both the offline and online EnKFs using the same variant of EnKFs, the results can be generalized to all classes of EnKFs. Furthermore, this paper adopted the LESTKF and the EnKF derivation in Section 2 applies to the LESTKF. Although we mentioned the work of Godinez and Moulton (2012) in Section 1, a section about the derivation of the matrix-free EnKF using Sherman–Morrison–Woodbury formulas seems to be beyond the scope of this paper. We believe it would be better to refer the interested readers to the paper of Godinez and Moulton (2012).

2. In section 2, I do not agree with the statement: *the Cholesky decomposition is more efficient than the SVD, but the SVD is more robust if the matrix is significantly ill-conditioned.* For unlocalized EnKF formulations, via the SVD decomposition, we can obtain EnKF implementations whose computational effort reads $O(n \cdot N^2)$, where $n$ is the number of model components and $N$ is the ensemble size. On the other hand, by employing a direct solution (i.e., by employing a Cholesky decomposition), we can get EnKF formulation whose analysis steps can be computed with $O(n^3)$ long computations.

Thank you for pointing out this possible misunderstanding. The statement: *The Cholesky decomposition is more efficient than the SVD, but the SVD is more robust if the matrix is significantly ill-conditioned.* is under the context of finding the square root of $\mathbf{I} - \mathbf{S}^T \mathbf{F}^{-1} \mathbf{S}$. It does not mean that the all-over performance of an EnKF using Cholesky decomposition is more efficient than the one using SVD. To avoid any possible misunderstanding, we deleted this statement from the manuscript (L19P6 to L21P6):

3. As you may know, **localization methods** are crucial to getting accurate analysis states. These methods mitigate the impact of spurious correlations in EnKF based formulations. In operational contexts, when domain decomposition is employed during assimilation steps, the dimension of local boxes can still be much larger than that of ensemble sizes. Therefore, local analysis increments can be poorly estimated at each sub-domain. How does your proposed method deal with this? I would like you to increase the discussion in this direction, you have already cited two papers that deal with this:

    (a) Anderson, J. L. (2001). An ensemble adjustment Kalman filter for data assimilation. Monthly weather review, 129(12), 2884-2903.

    (b) Nino-Ruiz, E. D., Sandu, A., & Deng, X. (2019). A parallel implementation of the ensemble Kalman filter based on modified Cholesky decomposition. Journal of Computational Science, 36, 100654.

Localization methods can effectively extenuate the impact of spurious correlations of the long-distance pairs. A local implementation based on domain localizations of EnKFs is very efficient and accurate for local observations, but has difficulties for non-local observations, especially for satellite measurements with long spatial correlations. For observations with long spatial correlation, the effective size of a local box would be significantly larger than the size of the ensemble, therefore this implication of the ensemble being too small for the local box could lead to a poor local analysis. Localization methods not only are crucial to the analysis accuracy by suppressing spurious correlations but also have a great impact on the computational efficiency. For example, a parallel implementation of EnKF based on modified Cholesky decomposition (Nino-Ruiz and Sandu, 2015, 2017; Nino-Ruiz et al., 2018, 2019) demonstrates an improvement of the analysis accuracy as the increasing of the influence radius, but the improved accuracy comes at the cost of increasing computations. On the other hand, the LETKF deteriorates the analysis accuracy as the increasing influence radius.

The main purpose of this study is to compare the computational efficiencies between offline and online EnKFs. Both offline and online EnKFs in this study use the same

localization to guarantee that the difference between the computational times comes only from the coupling ways: offline or online. Therefore, including a paragraph or section to discuss localizations seems to deviate from the central topic of this paper since this study does not investigate the accuracy or the efficiency of localizations. But we think a brief mention of the difficulties of localizations for local EnKFs for observations with long spatial correlations would be proper in the introduction section in the manuscript (L32P2 to L7P3):

 The local implementation based on domain localizations of EnKFs is very efficient and accurate for local observations, but has difficulties for non-local observations, especially for satellite measurements with long spatial correlations. For observations with long spatial correlation, the effective size of a local box would be significantly larger than the size of the ensemble, therefore this implication of the ensemble being too small for the local box could lead to a poor local analysis. Localization methods not only are crucial to the analysis accuracy by suppressing spurious correlations but also have a great impact on the computational efficiency. For example, a parallel  implementation of EnKF based on modified Cholesky decomposition  (Nino-Ruiz and Sandu, 2015, 2017; Nino-Ruiz et al., 2018, 2019)  demonstrates an improvement of the analysis accuracy as the increasing of the influence radius, but the improved accuracy comes at the cost of increasing computations. On the other hand, the LETKF deteriorates the analysis accuracy as the increasing influence radius.

4. While localization methods reduce the impact of spurious correlations, **covariance inflation** mitigates the impact of the under-estimation of sample variances. Most of the square-root based formulations (i.e., ETKF, and LETKF) suffer from under-estimation of sample variances, but, you do not mention this relevant topic in your entire paper.

We agree with the reviewer that covariance inflation can mitigate the underestimated covariance which is due to the finite size of the ensemble and model imperfections, therefore improve the ensemble spread. But bearing in mind, covariance inflation, especially empirical inflation, would lead to unbalanced ensemble members even with adaptive inflation. Because the main purpose of this study is to investigate the efficiencies between offline and online EnKFs, the multiplicative coefficient of covariance inflation is set to one to keep its computation, that is, covariance inflation has no effect but the computational time of covariance inflation is included. In other words, the results of the computational efficiencies between offline and online EnKFs in this study are still valid when covariance inflation takes effects. we added a sentence to describe the setting of covariance inflation of our experiments in the manuscript (L1P21 to L3P21):

The multiplicative coefficient of covariance inflation is set to one to keep its computation but has no effect on the covariance matrix so that the total computational time includes the similar computational time of covariance inflation whether it takes effects or not.

5. When domain decomposition is employed, it is very common to send boundary information to obtain physically consistent solutions, in this manner, solutions at different processors do not look like independent domains but, they are "*connected*" since neighboring sub-domains share boundary information. I may misunderstand but, are you

sharing boundary information among neighboring sub-domains? Thus far, it seems like not, right? If this is correct, how do you guaranty that global solutions (once all local analysis are mapped back onto the global domain) are consistent with the physics and the dynamics of numerical models?.

No, our implementation does not exchange the shared boundary information **during EnKF analyses** but does exchange the observations among processors **before EnKF analyses** so that each processor has its observations including those near the boundaries but within the influence radius of the localization and can do its own local analyses independently and in parallel. And the weights of the localization are calculated by a 5th order polynomial (Gaspari and Cohn, 1999) and decrease smoothly to zero as the influence radius increasing to the specified threshold to guarantee the continuity of the global analysis at boundaries of subdomains after all local analyses are mapped back onto the global domain. Thus, the global analysis is consistent with the physics and dynamics of the model in terms of subdomain boundary continuities. If you mean the real consistency of physics and dynamics of the model, in our point of view, EnKFs cannot guarantee a perfect physical and dynamical consistency after assimilations, even 4DVAR with dynamical constraints in its cost function cannot promise that, not to mention deficiencies of EnKFs due to the finite size of the ensemble which leads to the underestimated background covariance, certainly, this can be mitigated to a certain extent by a large size of ensembles. We detailed the weights of localizations that maintain the continuities between the subdomain boundaries in the manuscript (L30P20 to L33P20):

 which calculates the localization weights using a 5th order polynomial (Gaspari and Cohn, 1999)  Because localization weights decrease smoothly to zero as the influence radius increasing to the specified threshold, this fact guarantees the continuity of the global analysis at boundaries of subdomains after the local analyses are mapped back onto the global domain.

6. What observational network do you employ during experiments? are employing a full observational network (all model components are observed)? What is the performance of your method (in terms of accuracy) as the observational network becomes sparse?

As we explained in the answers to the first two reviewers (https://www.geosci-model-dev-discuss.net/gmd-2019-132/gmd-2019-132-AC3-supplement.pdf), a simple and idealized model and synthetic observations are adopted in this study to compare the efficiencies between offline and online EnKFs. That is only 10% of model components are randomly chosen to have an observation. The objective is to compare the efficiency but not the accuracy of a specific variant of EnKF methods. Many factors (such as the adopted specific variant of EnKF methods, its localization, covariance inflation if applicable, the sparsity of observations, the accuracy of observations, the nonlinearity of the model, just to name a few) could affect the ultimate accuracy of assimilations. The variant of EnKF methods we adopted is the LESTKF which is recommended in PDAF (Parallel Data Assimilation Framework at http://pdaf.awi.de), also the localization follows the recommended practice. Its accuracy would be similar to other variants of EnKF methods in the situation of sparse observations.

I hope these comments help to improve your paper.

**References**

[revised manuscript text omitted]